# An expanded toolkit for gene tagging based on MiMIC and scarless CRISPR tagging in *Drosophila*

**David Li-Kroeger[1†], Oguz Kanca[1†], Pei-Tseng Lee[1], Sierra Cowan[2], Michael T Lee[2], Manish Jaiswal[1,3‡], Jose Luis Salazar[1,4], Yuchun He[1,3], Zhongyuan Zuo[1], Hugo J Bellen[1,3,4,5,6]***

[1]Department of Molecular and Human Genetics, Baylor College of Medicine, Houston, United States; [2]Department of Biochemistry and Cell Biology, Rice University Houston, Houston, United States; [3]Howard Hughes Medical Institute, Baylor College of Medicine, Houston, United States; [4]Program in Developmental Biology, Baylor College of Medicine, Houston, United States; [5]Department of Neuroscience, Baylor College of Medicine, Houston, United States; [6]Jan and Dan Duncan Neurological Research Institute, Houston, United States

**Abstract** We generated two new genetic tools to efficiently tag genes in *Drosophila*. The first, Double Header (DH) utilizes intronic MiMIC/CRIMIC insertions to generate artificial exons for GFP mediated protein trapping or T2A-GAL4 gene trapping in vivo based on Cre recombinase to avoid embryo injections. DH significantly increases integration efficiency compared to previous strategies and faithfully reports the expression pattern of genes and proteins. The second technique targets genes lacking coding introns using a two-step cassette exchange. First, we replace the endogenous gene with an excisable compact dominant marker using CRISPR making a null allele. Second, the insertion is replaced with a protein::tag cassette. This sequential manipulation allows the generation of numerous tagged alleles or insertion of other DNA fragments that facilitates multiple downstream applications. Both techniques allow precise gene manipulation and facilitate detection of gene expression, protein localization and assessment of protein function, as well as numerous other applications.
DOI: https://doi.org/10.7554/eLife.38709.001

**\*For correspondence:**
hbellen@bcm.edu

[†]These authors contributed equally to this work

**Present address:** [‡]TIFR Centre for Interdisciplinary Sciences, Hyderabad, India

## Introduction

Comprehensive gene annotation is a central challenge in the post-genomic era. *Drosophila melanogaster* offers more sophisticated genetic approaches and tools to assess gene function and expression than other multicellular model organisms (*Bier et al., 2018*; *Cox et al., 2017*; *Germani et al., 2018*; *Heigwer et al., 2018*; *Kanca et al., 2017*; *Kaufman, 2017*; *Simpson and Looger, 2018*). A versatile tool used for functional gene annotation in *Drosophila* is MiMIC, a *Minos*-based transposon that integrates a Swappable Integration Cassette (SIC) in the genome (*Venken et al., 2011a*). MiMIC SICs contain a cassette nested between two *attP* sites that can be exchanged with any DNA sequence flanked with *attB* sites through Recombinase Mediated Cassette Exchange (RMCE) by ΦC31 integrase. When a MiMIC is integrated in an intron of a gene flanked on both sides by coding exons (hereafter referred to as a coding intron), the SIC can easily be exchanged with an artificial exon that encodes *Splice Acceptor (SA)-(GGS)$_4$ linker-EGFP-FlAsH tag-StrepII tag-TEV protease cleavage site-3XFlag-(GGS)$_4$ linker-Splice Donor (SD)* (abbreviated as *GFP tag*) (*Venken et al., 2011a*; *Nagarkar-Jaiswal et al., 2015a*; *Nagarkar-Jaiswal et al., 2015b*). The GFP-tagged endogenous proteins report the subcellular localization of the gene product and are

**eLife digest** Organisms have tens of thousands of genes, but finding out exactly what they all do is one of the greatest challenges of modern genetics. To understand a gene's job, it's necessary to find out what gene is active in which tissue, where their proteins are located within the cell, and what happens when the sequence of a gene is altered or removed. This multi-step process of 'annotating' genes can be challenging in practice.

One common approach is to make use of a DNA pattern called a MiMIC and insert it in a specific part of the gene called an intron. A tag for a protein that glows green under the microscope can then be added to a MiMIC to help visualize where and when the protein is being expressed. MiMICs can also be used to integrate a system called T2A-GAL4, which typically creates a severe mutation in the gene and allows to track the timing of when and where the gene is expressed. This helps to discover the role of the gene in cells and tissues. However, a problem with this approach is that when either the protein tag or the T2A-GAL4 system is added, half of the time they point into the wrong direction. This is because each DNA strand is read in one direction only.

Now, Li-Kroeger et al. created a so-called 'Double Header' system, which includes T2A-GAL4 coding in one direction and the protein tag in the other. Therefore, when the system integrates, there will always be one tag pointing in the correct direction. This makes the system twice as efficient.

Not all genes have introns though. To access genes that do not contain introns, Li-Kroeger et al. developed another system, which uses the genome editing tool CRISPR-Cas9 to introduce a different kind of visible marker. Here, the whole gene is typically removed and replaced by a visible marker, which can then be replaced by any DNA, including protein tags and the T2A-GAL4 system.

With these approaches, all genes in the fruit fly can now be targeted. The systems perform several tasks, including detecting gene activity and the location of proteins in the cell, and analyzing the role of the protein. The findings will be relevant to researchers interested in fruit fly genetics and cell function.

DOI: https://doi.org/10.7554/eLife.38709.002

---

functional in ~75% of tested genes (*Nagarkar-Jaiswal et al., 2015a*). Importantly functional GFP-protein traps can be used for multiple assays. These include chromatin Immunoprecipitation (ChIP) of transcription factors (*Nègre et al., 2011*), Immunoprecipitation (IP)-Mass Spectroscopy (MS) (*David-Morrison et al., 2016*; *Neumüller et al., 2012*; *Yoon et al., 2017*), rapid conditional removal of gene products (*Caussinus et al., 2011*; *Lee et al., 2018b*; *Nagarkar-Jaiswal et al., 2015a*; *Neumüller et al., 2012*; *Wissel et al., 2016*) and sequestration of tagged proteins (*Harmansa et al., 2017*; *Harmansa et al., 2015*). Hence, tagging an endogenous gene with GFP enables numerous applications to dissect gene function.

The SIC in MiMICs can also be replaced by an artificial exon that encodes *SA-T2A-GAL4-polyA signal* (abbreviated as *T2A-GAL4*) (*Diao et al., 2015*; *Gnerer et al., 2015*; *Lee et al., 2018a*). *T2A-GAL4* creates a mutant allele by truncating the protein at the insertion site but also expresses GAL4 with the spatial-temporal dynamics of the targeted gene. Hence, *T2A-GAL4* facilitates the replacement of the gene of interest with fly or human UAS-cDNAs (*Bellen and Yamamoto, 2015*; *Şentürk and Bellen, 2018*; *Wangler et al., 2017*; *Lee et al., 2018a*), allowing one to assess putative disease-associated variants and permitting structure-function analysis of the protein of interest. Moreover, these gene-specific GAL4 stocks can be used to drive a variety of UAS constructs to further identify and probe the function of the cells expressing the gene using UAS-Fluorescent proteins or numerous other UAS constructs (*Venken et al., 2011b*). This is especially useful for genes that are not abundantly expressed, providing a means to amplify the signal, as GAL4 drives overexpression of the UAS transgenes (*Diao et al., 2015*; *Lee et al., 2018a*). In summary, MiMIC applications allow the acquisition of valuable data about the function of the gene as well as the cells in which the gene is expressed.

Given the usefulness of MiMICs, the *Drosophila* Gene Disruption Project (GDP) (http://flypush. imgen.bcm.tmc.edu/pscreen) has generated and mapped 17,500 MiMIC insertion stocks (*Nagarkar-Jaiswal et al., 2015a*; *Venken et al., 2011a*). This collection includes insertions within introns

for ~1860 genes, each of which can be converted to a *GFP*-tagged protein trap and/or a *T2A-GAL4* gene trap (*Nagarkar-Jaiswal et al., 2015a*; *Nagarkar-Jaiswal et al., 2015b*; *Lee et al., 2018a*). However, we needed to develop a complementary strategy to generate resources for genes that do not have a MiMIC randomly inserted within a coding intron. To that end, we recently developed CRIMIC (CRISPR mediated Integration Cassette); a Cas9/CRISPR Homology Directed Repair (HDR) mediated approach that integrates a modified SIC (*attP-FRT-SA-T2A-GAL4-polyA-FRT-attP*) in a coding intron of choice. This approach greatly expands the number of genes that can be tagged using MiMIC-like technology from 1860 to ~6000 (*Lee et al., 2018a*) allowing about forty percent of *Drosophila* protein coding genes to be targeted with SICs.

RMCE cassettes can either be injected into embryos as part of a circular plasmid or can be circularized in vivo from an initial insertion locus in the genome through Cre/loxP or Flp/FRT mediated recombination (*Diao et al., 2015*; *Nagarkar-Jaiswal et al., 2015b*). Importantly, RMCE cassettes can replace a SIC in either orientation with equal probability due to inverted symmetric *attP* sequences. Therefore 50% of the insertions are inserted in the opposite orientation of transcription and will not be included in the mature mRNA. Hence, only half of all successful exchange events will result in protein or gene trap lines.

Here, we show that by combining GFP-protein traps and T2A-GAL4 gene traps in a single RMCE construct, named Double Header (DH), we significantly increased the number of productive RMCE events for MiMIC/CRIMIC containing genes to generate protein or gene trap alleles. Importantly, we expand the ability to target SICs into genes regardless of the presence of introns to allow access to virtually any gene in the fly genome based on CRISPR/Cas9-mediated HDR. This provides a means to create robust null alleles with simple screening, and to convert the SIC insertion using any DNA, creating scarless modifications to facilitate numerous downstream applications.

## Results

### Double Header (DH) improves the tagging rate of MiMIC containing genes

SICs in coding introns can be converted into GFP-protein traps or T2A-GAL4 gene traps through RMCE. However, because RMCE of SICs in MiMICs and CRIMICs can occur in either orientation, only one out of two events produces a tag that is incorporated in the gene product. Moreover, each RMCE experiment generates only a protein trap or a gene trap, requiring two independent injections or crosses to generate both reagents. In order to reduce the effort to generate these genetic tools we engineered Double Header (DH), a construct that combines the two key RMCE cassettes to replace intronic MiMICs and CRIMICs: (1) *SA-T2A-GAL4-polyA* (*DH^{T2A-GAL4}*) which generates a gene trap that expresses GAL4 in the expression domain of the gene while typically inactivating gene function, in one orientation and (2) *SA-GFP-SD* (*DH^{GFP}*) which generates a GFP-protein trap, in the opposite orientation. Hence, insertion of DH via RMCE in a MiMIC in either orientation should result in two valuable reagents. This compound RMCE cassette is flanked by two inverted φC31 *attB* sites in a vector backbone that contains other features including mini-*white* as shown in *Figure 1A*. The presence of *white* in the plasmid backbone allows for a counter selection against the integration of the whole plasmid when incorporated into a white⁻ background, ensuring that only the DNA between the *attB* sites integrates.

The artificial exons that are integrated into MiMIC or CRIMIC sites need to be in frame with the preceding Splice Donor (SD) to create a functional tag. Because exon/intron boundaries can occur at any one of three positions in a codon (phase 0,+1,+2), we generated three different DH plasmids (Sequences can be found in *Supplementary file 1*). Each construct contains the same codon phase for the two modules. Given that the RMCE cassette is about 4.8 kB, or about 1.5 times the size of the single T2A-GAL4 cassette, we anticipated a lower integration rate. We tested the integration efficacy by injecting 30 strains carrying a coding intronic MiMIC insertion. On average,~400 embryos were injected for each MiMIC line together with a plasmid that encodes the ΦC31 integrase. We screened for the loss of yellow+ in the progeny of the injected animals, as MiMICs carry the *yellow+* marker (*Figure 1—figure supplement 1A*). We did not observe the integration of *white* indicating that a single RMCE event leading to the integration of the entire plasmid is rare. For 16 out of 30 MiMICs, we were able to isolate yellow⁻ flies that carry a DH integration (*Figure 1B*; *Figure 1—*

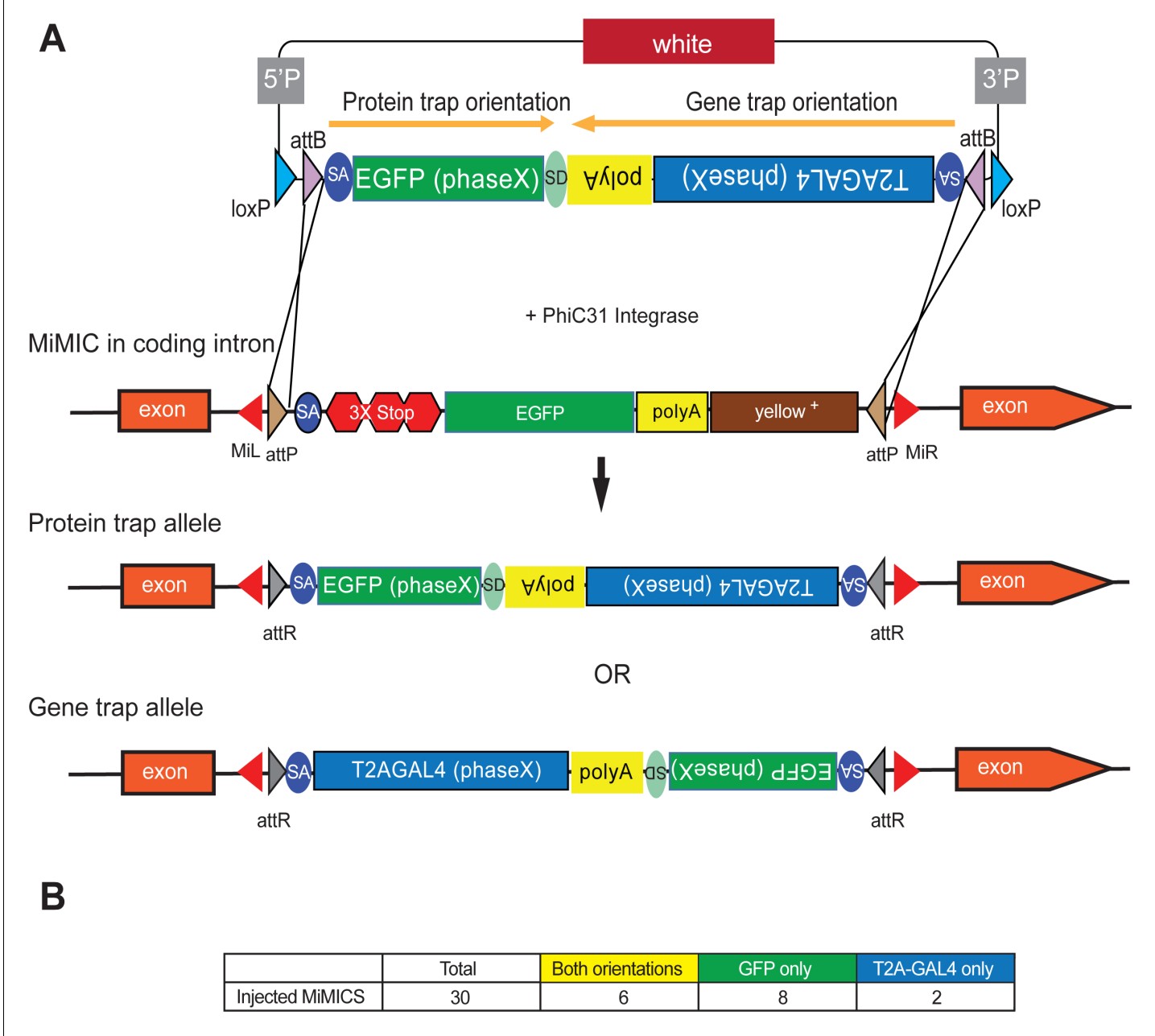

**Figure 1.** Double Header optimizes RMCE outcome of MiMICs. (**A**) Schematics of the Double Header construct and RMCE outcomes. Double Header constructs contain a Splice Acceptor (SA)- super folder GFP-FlAsH-StrepII-TEV-3xFlag (EGFP) – Splice Donor (SD) in one orientation and a SA-T2A-GAL4-polyA in the other orientation. Insertion in the GFP orientation results in GFP protein trap whereas insertion in the T2A-GAL4 orientation results in T2A-GAL4 gene trap. (**B**) Double Header injection statistics.

DOI: https://doi.org/10.7554/eLife.38709.003

The following figure supplements are available for figure 1:

**Figure supplement 1.** Injection data for Double Header.

DOI: https://doi.org/10.7554/eLife.38709.004

**Figure supplement 2.** PCR strategy to identify Double Header orientation.

DOI: https://doi.org/10.7554/eLife.38709.005

*figure supplement 1B*). We determined the orientation of DH inserts through single fly PCR (*Figure 1—figure supplement 2*) and determined the orientation of the cassette in 47 out of 72 DH RMCE events. For 6 MiMICs we obtained both orientations and for the other 10 MiMICs we obtained one or the other orientation (*Figure 1B*; *Figure 1—figure supplement 1B*). Hence, we generated a total of 22 new reagents, increasing the overall rate of productive RMCE events by injection (*Nagarkar-Jaiswal et al., 2015a*). Note that the integration efficiency of DH construct by injection is lower than the smaller SICs:~50% versus~66% (*Nagarkar-Jaiswal et al., 2015a*). However, the number of productive events increases tagging efficacy somewhat as every successful event produces a useful line: 74% versus 66%.

## In vivo double header mobilization based on transgenes

In an effort to avoid embryo injections and to increase integration efficiency of DH, we developed an in vivo RMCE strategy using genetic crosses similar to Trojan exons developed for *T2A-GAL4* (*Diao et al., 2015*). We integrated the same constructs as in *Figure 1A*, one for each reading frame, in the genome through P-element mediated transformation. These insertions serve as jump-starter constructs because the RMCE cassettes in these transgenes can be excised from their initial landing sites by expressing Cre recombinase in germ line cells (*Figure 2*). The crossing scheme is outlined in Figure 2-figure supplement 1. *Figure 3A*. We generated jump-starter insertions in second and third chromosomes for all three possible phases of DH and generated double balanced stocks for subsequent crosses. We tested the efficacy of integration by crosses for DH for third chromosome MiMICs. For 12 out of 13 MiMICs tested we obtained integration of DH. We determined the orientation of DH for 48 out of 102 insertions by PCR. The inconclusive insertions either showed no PCR amplification in one end or both ends of the MiMIC (48/102) or in rare cases conflicting PCR amplification that indicates integration in both orientations (6/102). Interestingly 44/54 inconclusive inserts happened in only two of the MiMICs, indicating locus specific issues. For 6 out of 12 MiMICs we obtained both orientations and for six we obtained one or the other, resulting in 18 tagged genes (*Figure 2—figure supplement 1*). Hence, the genetic strategy is about twice as efficient (18 constructs for 13 crossed MiMICs versus 22 constructs for 30 injected MiMICs) as the injection strategy in generating RMCE events and requires significantly less effort.

## Double Header reports the expression pattern of the tagged gene and protein

We proceeded to test whether DH functions as expected. We determined the expression patterns of genes tagged in both orientations in third instar larval brain and adult brains for *MI01487* (*kibra*), *MI05208* [*5-HT2B(5-hydroxytryptamine receptor 2B)*], *MI06794* [*Lgr4(Leucine-rich repeat-containing G protein-coupled receptor 4)*], *MI06872* (*CG34383*), *MI08614* [*Dgk (Diacyl glycerol kinase)*], *MI11741* (*CG12206*) and *MI15073* (*CG9132*) (*Figure 3*). As we selected a few MiMICs that were previously tagged with *T2A-GAL4* by the Gene Disruption Project as positive controls (*Lee et al., 2018a*; *Diao et al., 2015*) (*MI01487* (*kibra*), *MI06794* (*Lgr4*), *MI06872* (*CG34383*), *MI08614* (*Dgk*), and *MI11741* (*CG12206*), we were able to compare expression patterns obtained by $DH^{T2A-GAL4}$ to expression patterns obtained by single *T2A-GAL4* (http://flypush.imgen.bcm.tmc.edu/pscreen/rmce/). In all cases the expression pattern was very similar to what was previously reported (*Figure 3*) (*Lee et al., 2018a*). In addition, tracheal expression of *CG12206* is consistent with a previous report (*Chandran et al., 2014*) and the $5HT2B^{T2A-GAL4}$ expression pattern in the adult brain matches an independently generated T2A-GAL4 (*Gnerer et al., 2015*) (*Figure 3—figure supplement 1*). In all cases, the $DH^{GFP}$ insertions show consistent patterns of expression in third instar larval brains, albeit at much lower levels than the $DH^{T2A-GAL4}$ insertions at the same MiMIC site (*Figure 3*). Note that in adult brains almost no signal of $DH^{GFP}$ was detected, in agreement with previous observations, with the exception of *MI15073* which shows ubiquitous expression, (*Diao et al., 2015*; *Lee et al., 2018a*) (*Figure 3—figure supplement 1*). These results indicate that neither the size nor the design of DH alters the functionality or expression patterns of the tagged genes and proteins.

As the GFP protein traps should be able to report the subcellular localization of the tagged protein we turned to tissues where subcellular localization and specific cell expression is easily assessed. We therefore dissected and stained egg chambers with anti-GFP. We were easily able to visualize the GFP tagged proteins in the seven protein traps previously examined (*Figure 4*). The tagged

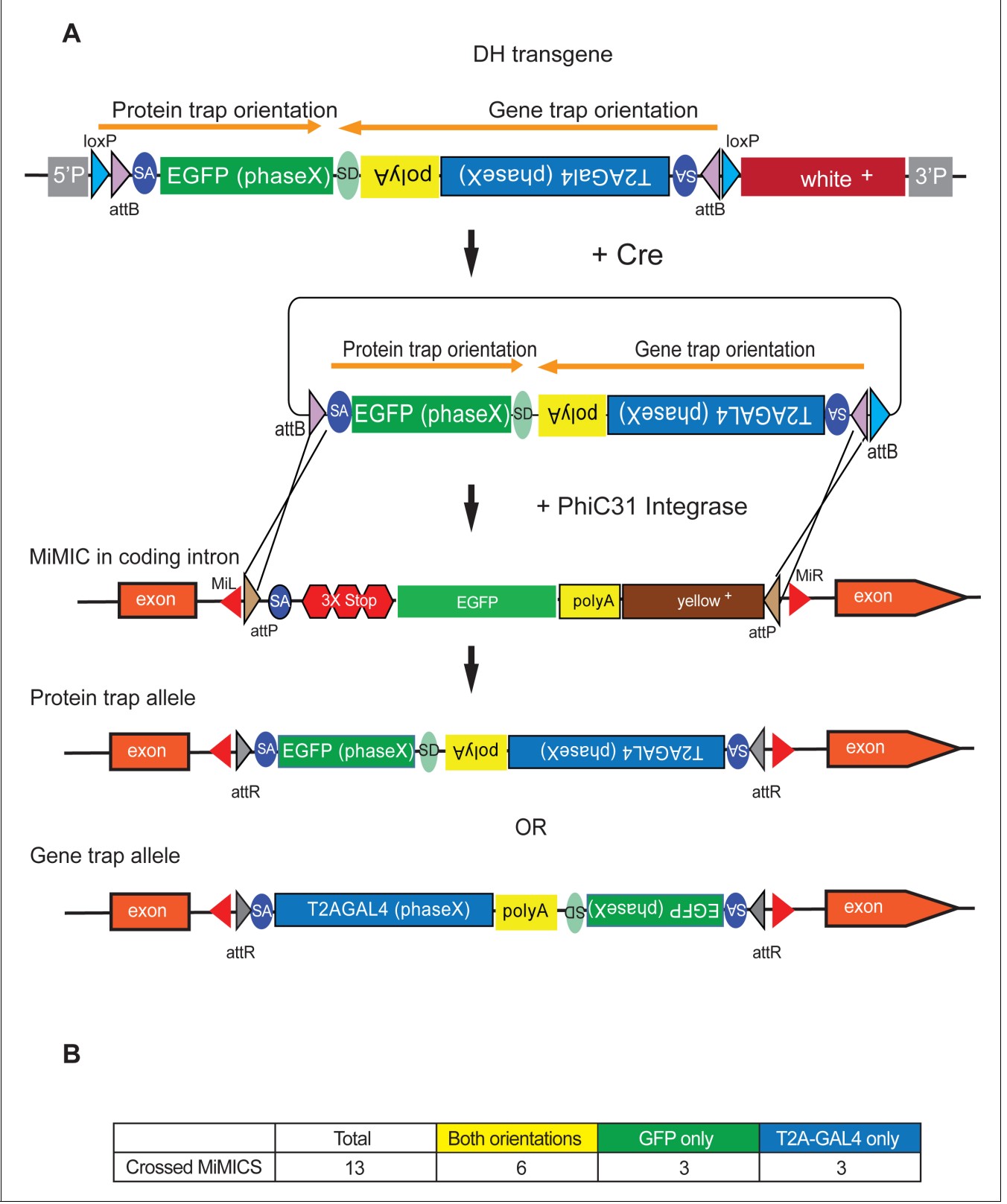

**Figure 2.** Double Header integration through crosses facilitates RMCE. (**A**) Schematics of the Double Header transgene mobilization in vivo. Double Header transgenes contain *loxP* sites that can be used to mobilize the RMCE cassette in vivo, without the need for injection. (**B**) Double header crossing statistics.

DOI: https://doi.org/10.7554/eLife.38709.006

*Figure 2 continued on next page*

*Figure 2 continued*

The following figure supplement is available for figure 2:

**Figure supplement 1.** Crossing scheme for Double Header and data of integration.
DOI: https://doi.org/10.7554/eLife.38709.007

proteins are shown in green and the nuclei are stained with DAPI in red. Kibra is detected in somatic follicle cell cytoplasm, including some migratory border cells. 5HT2B is expressed in both somatic and germline cells including the oocyte. Lgr4 is localized to germline nurse cell nuclei and is enriched in the oocyte anterior-dorsal and ventral cortex. CG34383 is mostly present in follicle cells, especially in their apical domain. Dgk is observed in nurse cell and follicle cell nuclei as well as their cytoplasm. CG12206 is quite enriched in the cytoplasm of centripetal cells and CG9131 is present in both germ cells and follicle cells and enriched in polar cells. In summary, GFP protein tagging with DH can be used to determine the cellular and subcellular localization of tagged proteins.

## A compact cassette to target a SIC to intron-less genes

To tag a gene containing a MiMIC/CRIMIC, the SIC should be integrated within a suitable coding intron, leaving about 50–60% of all *Drosophila* genes that encode proteins inaccessible. Targeting genes without introns by directly fusing tags in the proper reading frame has been very difficult because HDR is much less efficient than non-homologous end joining (NHEJ) (*Gratz et al., 2014*). Hence, expanding the range of targetable genes requires precise, seamless genome editing. Gene editing using HDR is well suited to modifying genes without introducing unwanted changes (*Bier et al., 2018*). HDR repairs double stranded DNA breaks using a donor template that contains two homology regions, each typically about 1000 nucleotides, which flank the desired changes. Recombination on either side of the break replaces the regions with the template, precisely modifying the locus. We therefore developed a novel SIC compatible with HDR (*Figure 5*; *Figure 5—figure supplements 1* and *2*) that could be targeted to loci regardless of the presence of introns.

To make a SIC that is HDR compatible, three features are important: (1) a dominant marker for screening that is compact for ease of insertion via HDR (*Li et al., 2014*), (2) a method to insert the SIC in the desired location, and (3) a strategy to remove the SIC for replacement with the desired end product. To design a compact marker that is compatible with Golden Gate cloning we focused on the *yellow* gene which has well characterized enhancers (*Geyer and Corces, 1987*). We identified a 575 nucleotide regulatory region that when fused to the promoter and *yellow* coding sequence creates a 2.9 kilobase cassette that reliably drives expression only in the wing (*Figure 5—figure supplement 2*). We refer to this cassette as $y^{wing2+}$.

To enable targeting $y^{wing2+}$ into precise locations in the genome, we first define a region (or gene) of interest (ROI) flanked by two Cas9 target sites comprised of a 20 nucleotide guide sequence and an NGG PAM (*Jinek et al., 2012*; *Sternberg et al., 2014*) (*Figure 5A*). We then design a HDR donor template with $y^{wing2+}$ flanked by homology regions. As the donor template removes part of the Cas9 target sequences neither the donor cassette nor the final product are cleaved upon HDR. Injecting the donor template along with sgRNA expression plasmids into embryos carrying a germline-specific source of Cas9, followed by screening offspring for yellow + wings provides a straightforward method to generate robust null alleles for the gene.

Lastly, to make the cassette removable, we flanked the SIC with the nucleotides 'GG' and 'CC' upstream and downstream of the $y^{wing2+}$ marker, respectively. Upon insertion, this creates two novel Cas9 target sites that are not present in the endogenous sequence (box inset of *Figure 5A*), which can be used to remove the inserted cassette for final replacement via a second round of HDR. Finally, to facilitate cloning, the $y^{wing2+}$ cassette was made compatible with Golden Gate assembly (*Engler et al., 2009*). We also generated templates for designing replacement HDR constructs containing GFP and mCherry for protein tags or T2A-Gal4 that are compatible with Golden Gate assembly (*Figure 5—figure supplement 1*).

## The $y^{wing2+}$ SIC efficiently replaces genomic loci

We tested the efficacy of replacing the coding sequence of 10 loci with $y^{wing2+}$ (*Table 1*). Nine out of ten injections led to successful integration of the cassette. We injected an average of ~500

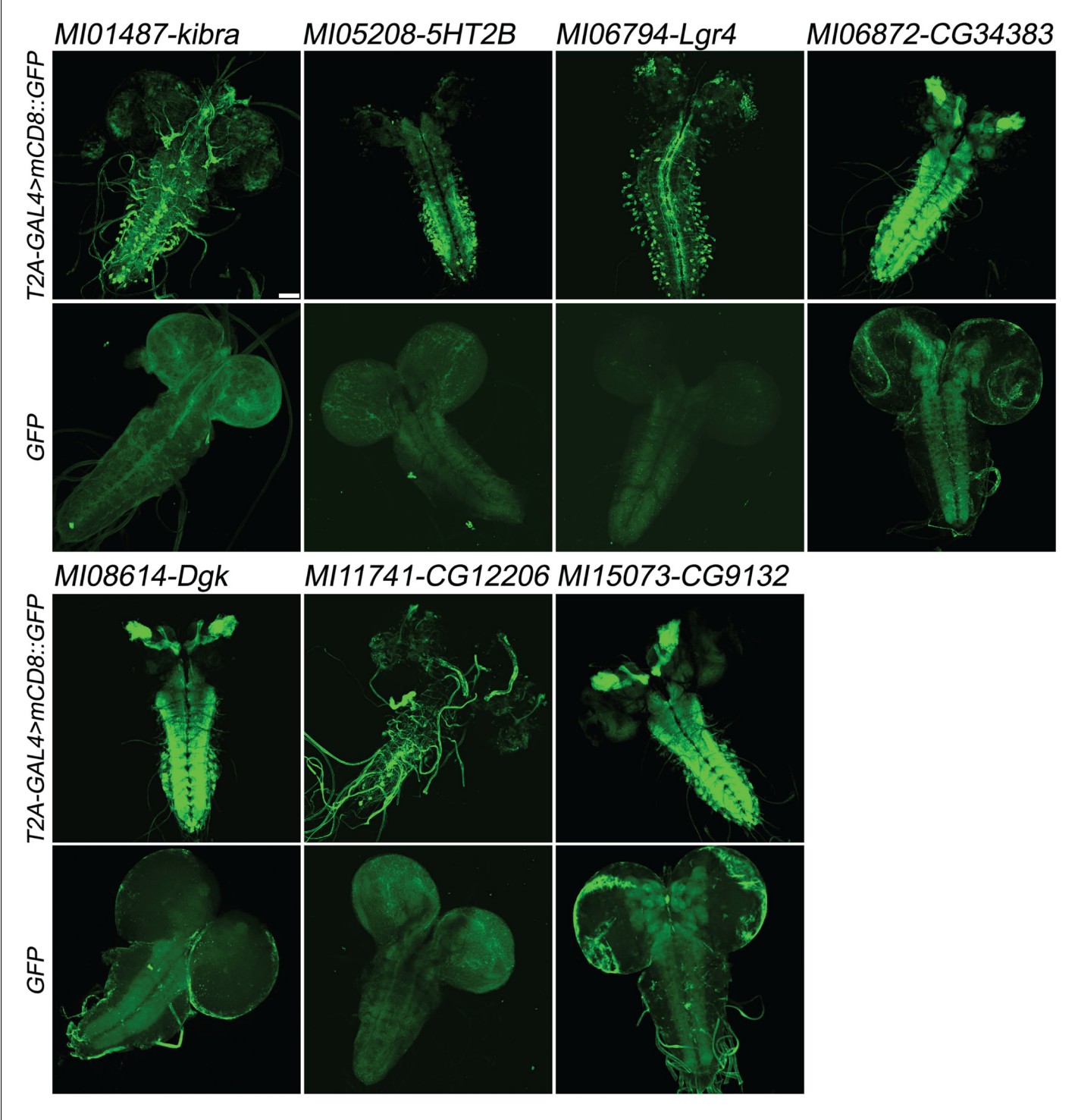

**Figure 3.** Examples of gene expression patterns obtained by Double Header. Each MiMIC, *MI01487*, *MI05208*, *MI06794*, MI06872, *MI08614*, *MI11741* and *MI15073*, was converted to either T2A-GAL4 protein traps or GFP protein traps by Double Header insertion. The expression in the larval CNS is shown with either *T2A-GAL4 > UAS-mCD8::GFP* or *GFP-tag* (GFP and mCD8::GFP, green). The affected genes are labelled above. Scale bar: 50 μm.
DOI: https://doi.org/10.7554/eLife.38709.008

The following figure supplement is available for figure 3:

**Figure supplement 1.** Examples of gene expression patterns obtained by Double Header insertions in MiMICs in adult brain.
DOI: https://doi.org/10.7554/eLife.38709.009

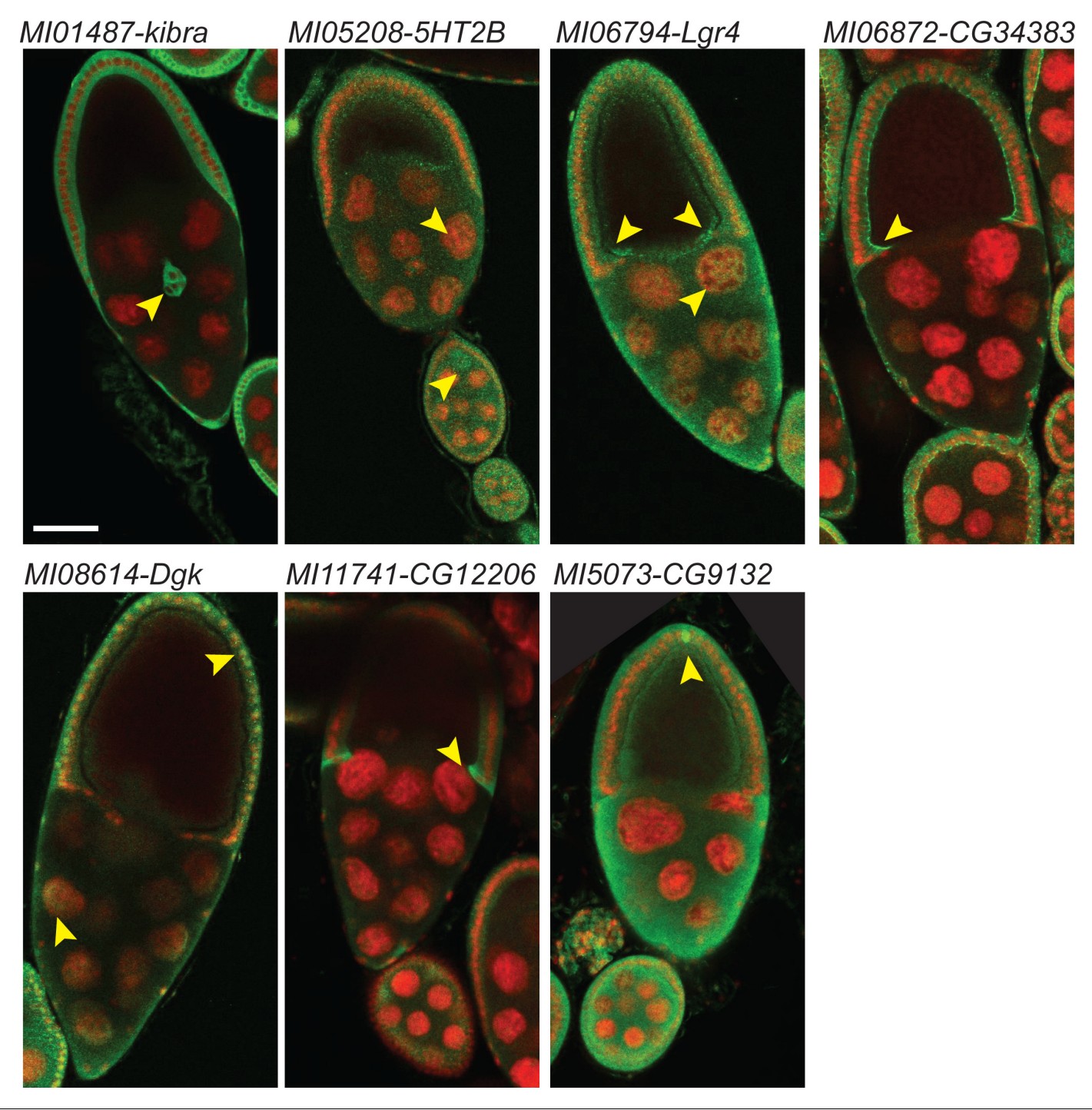

**Figure 4.** Examples of cellular expression patterns and subcellular localization of tagged proteins in egg chambers at stage 9 and 10. Double header GFP protein traps of MIMIC lines shown in *Figure 3* were dissected and ovaries were stained with anti-GFP antibody (green) and DAPI (red). Arrowheads indicate features that are referred to in the text; border cells for *kibra*; nurse cells, follicle cels and oocytes for 5HT2B; GFP is broadly expressed and distributed for Lgr4; note the apical enrichment in follicle cells in CG34383; nuclear and cytoplasmic staining in nurse cells and follicle cells are observed in Dgk; centripedal cells cytoplasm is mostly labeled in CG12206; broad expression and localization with pole cell enrichment in CG9132. Scale bar: 50 μm.

DOI: https://doi.org/10.7554/eLife.38709.010

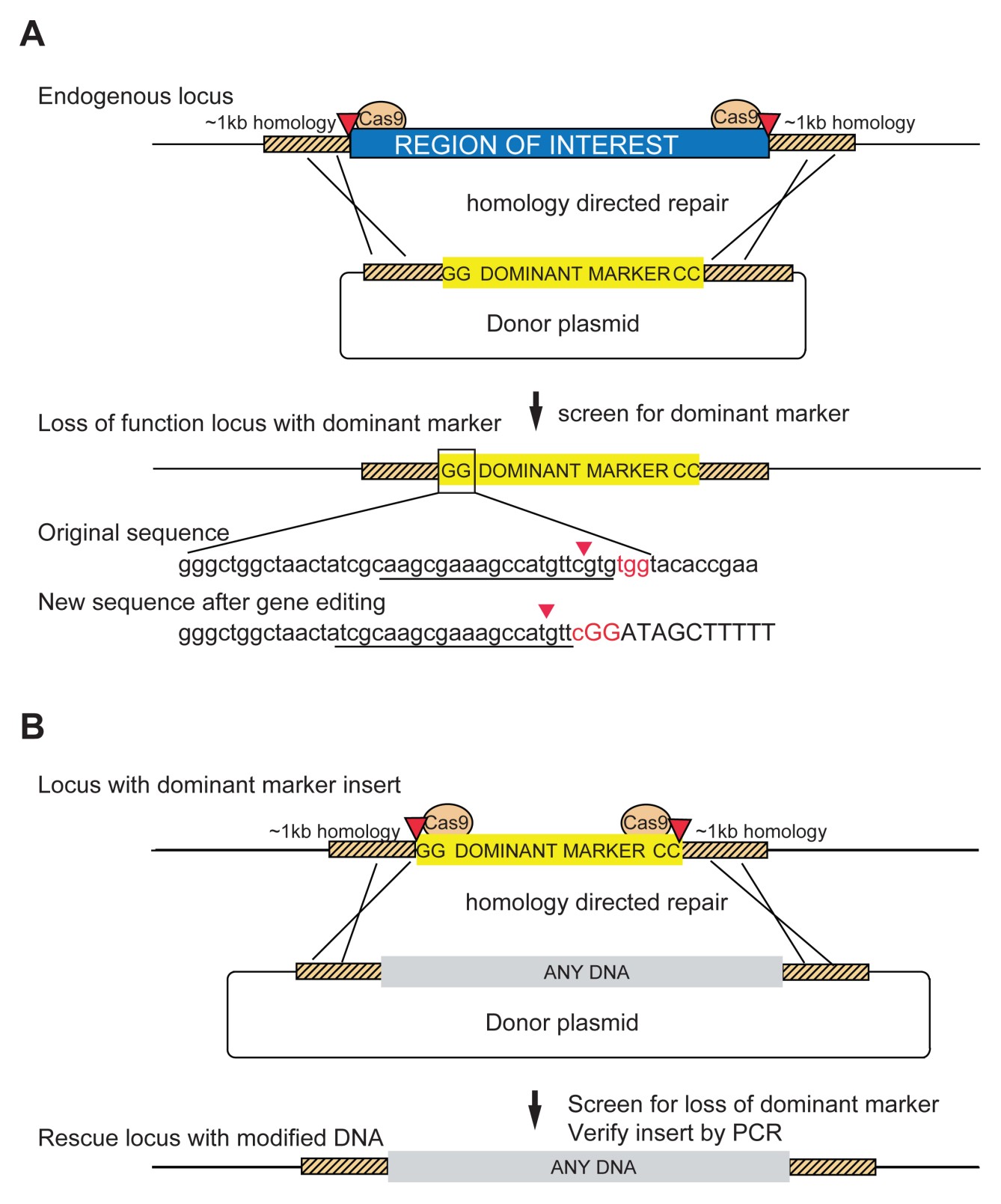

**Figure 5.** Schematic of a two-step system for scarless gene editing. (**A**) In step 1, a cassette containing a dominant marker flanked by nucleotides GG and CC replaces an endogenous locus via Homology Directed Repair (HDR) following double strand breaks produced by Cas9 cleavage (marked by red arrowheads). The removal of the intervening sequence between the Cas9 cut sites alters the sgRNA target sequences (underlined) preventing cleavage of the donor construct or the modified DNA. Screening for the dominant marker facilitates identification of CRISPR gene editing events while

*Figure 5 continued on next page*

*Figure 5 continued*

the flanking nucleotides GG (boxed inset) and CC create novel Cas9 target sites, allowing subsequent excision. **(B)** In step two the insert is removed and replaced with any DNA via a second round of HDR with new sgRNA sequences, facilitating the scarless insertion of any DNA sequence desirable.

DOI: https://doi.org/10.7554/eLife.38709.011

The following figure supplements are available for figure 5:

**Figure supplement 1.** Template vectors for cloning *yellow* expression constructs.

DOI: https://doi.org/10.7554/eLife.38709.012

**Figure supplement 2.** Mapping cis-regulatory modules for the *yellow* gene.

DOI: https://doi.org/10.7554/eLife.38709.013

embryos for each gene and recovered 1 to 6 independent founder lines for a total of 22 insertion events or ~2 founder animals/gene. Sanger sequencing confirmed the correct insertion of all but one HDR event, in which the entire plasmid backbone had been integrated. As shown in *Table 1*, seven insertions are homozygous lethal. To test whether the lethality was specific to the removal of the targeted gene, we attempted rescue of the lethality with a genomic duplication of the locus for four genes (*Nmnat (Nicotinamide mononucleotide adenylyltransferase), CG13390, Med27 (Mediator complex subunit 27), and CG11679,* and tested for failure to complement molecularly defined deletions for two genes (*ubiquilin* and *Nmnat*) (*Zhai et al., 2006*). In all cases, lethality mapped to the targeted locus showing that no second-site lethal mutations were induced in these lines. For the gene almondex (amx), the y$^{wing2+}$ insertion produced flies that were female sterile. Female sterility was previously observed for *amx* and a genomic fragment previously reported to rescue female sterility likewise rescued this phenotype in amx$^{\Delta CDS,ywing2+}$(*Cohorts for Heart and Aging Research in Genomic Epidemiology consortium et al., 2016*). For the gene *Stub1 (STIP1 homology and U-box containing protein 1)*, four positive lines were recovered; two are homozygous lethal while two are viable and fertile. Sanger sequencing confirmed the correct insertion of the cassette in all four lines, suggesting that the gene is not essential. Hence, the lethality is either caused by off-target cleavage events or the presence of a floating lethal mutation in the original strain. Thus while off-target cleavage may have occurred, this evidence suggests that it is not common, in agreement with what we

**Table 1.** Summary statistics for cassette knock-in experiments

| Construct | Genotype injected: | No. independent positive lines obtained | Lethality | Rescue of lethality/failure to complement |
|---|---|---|---|---|
| y$^{wing2+}$ ΔNmnat | y$^1$ M{nos-Cas9.P}ZH-2A w* | 6 | lethal | Genomic Fragment (*Zhai et al., 2006*)/ NmnatΔ4790–1 |
| y$^{wing2+}$ ΔStub1 | y$^1$ M{nos-Cas9.P}ZH-2A w* | 4 | Viable/ Fertile* | ND |
| y$^{wing2+}$ ΔUbqn | yw iso#6; +/+; attP2(y-){nos-Cas9} | 2 | lethal | Fails to complement Ubqn1 |
| y$^{wing2+}$ ΔItp-r83A | y$^1$ M{nos-Cas9.P}ZH-2A w* | 1 | lethal | ND |
| y$^{wing2+}$ ΔCG18769 | y$^1$ M{nos-Cas9.P}ZH-2A w* | 2 | lethal | ND |
| y$^{wing2+}$ ΔCG13390 | y$^1$ M{nos-Cas9.P}ZH-2A w* | 2 | lethal | Rescued by Genomic Fragment (this study) |
| y$^{wing2+}$ ΔMed27 | yw iso#6; +/+; attP2(y-){nos-Cas9} | 1 | lethal | Rescued by Genomic Fragment (this study) |
| y$^{wing2+}$ ΔCG11679 | yw iso#6; +/+; attP2(y-){nos-Cas9} | 2 | lethal | Rescued by genomic duplication BSC Dp(1:3) 304 |
| Y$^{wing2+}$ Δrho | y$^1$ M{nos-Cas9.P}ZH-2A w* | 0 | NA | NA |
| y$^{wing2+}$ Δamx | yw iso#6; +/+; attP2(y-){nos-Cas9} | 2 | Female sterile | Rescued by Genomic Fragment |

*two of four lines

DOI: https://doi.org/10.7554/eLife.38709.014

have observed when we use CRIMIC (*Lee et al., 2018a*). In summary, we created null alleles for nine genes and show that the $y^{wing2+}$ knock in cassette inserted precisely based on Sanger sequencing.

## Step two: removal of $y^{wing2+}$ allows 'scarless' modification of endogenous loci

The $y^{wing2+}$ cassette is designed to introduce two new gRNA target sites upon replacing the endogenous locus. These newly introduced gRNA target sites can now be used for the removal of the cassette via CRISPR/Cas9 mediated HDR and replacement with the desired DNA sequence. We attempted to incorporate protein tags for five genes (*Table 2*). We successfully incorporated tags for *Nmnat, Stub1, CG11679* and *Med27* but failed for *amx*. We tagged *Nmnat* and *Stub1* with GFP, and *CG11679* and *Med27* with Flag tags (see Mat. and Meth.). Internally GFP-tagged *Nmnat* (*Nmnat::GFP::Nmnat)* and C-terminally Flag tagged *CG11679* (*CG11679::Flag*) reverted the lethality of the $y^{wing2+}$ knock in allele and hence produced functional proteins. However, the C-terminal Flag-tagged *Med27* (*Med27::Flag*) is recessive pupal lethal, similar to the $y^{wing2+}$ knock in allele, suggesting that the C-terminal Flag tag disrupts protein function. Because the loss of *Stub1* (*Table 1*) does not result in an overt phenotype, we cannot determine if *Stub1::GFP* is functional but Sanger sequencing showed that the replacement of $y^{wing2+}$ with *Stub1::GFP* happened precisely. Taken together, the data indicate that Cas9 mediated cassette replacement occurred correctly for four out of five genes.

## A case study: structure function analysis of *NMNAT*

To highlight the utility of the $y^{wing2+}$ scarless replacement strategy, we performed a structure-function analysis of *Nmnat*. Nmnat is an enzyme with Nicotinamide adenine dinucleotide (NAD) synthase activity that also functions as a molecular chaperone (*Zhai et al., 2006*; *2008*). Additionally, the Nmnat family is highly conserved, required for neuronal survival and protects neurons from a variety of neurodegenerative insults (*Ali et al., 2013*; *Brazill et al., 2017*; *Lau et al., 2009*). A previous report generated a *Nmnat* null allele and determined that its loss causes lethality in first instar larva (*Zhai et al., 2006*). For our structure-function analysis, we first created a null allele of *Nmnat* by replacing the entire coding sequence (CDS) with $y^{wing2+}$, $Nmnat^{\Delta CDS,ywing2+}$ (see *Table 1*). The resulting flies are homozygous first instar larval lethal, consistent with a known protein null, and can be rescued by a 3 kb Nmnat transgene known to rescue the lethality associated with the loss of Nmnat

**Table 2.** Summary statistics for cassette swapping experiments

| Construct | Injected genotype: | No. embryos injected | No. fertile adults | No vials with y- flies | % of y- flies confirmed positive |
|---|---|---|---|---|---|
| Nmnat:GFP:Nmnat $^{wt}$ #1 | $y1$ $M\{nos\text{-}Cas9.P\}ZH\text{-}2A$ $w^*$;+; $y^{wing2+}$ $\Delta Nmnat/$ $TM6B$ | 514 | 7 | 4 | 6% |
| Nmnat:GFP:Nmnat $^{wt}$ #2 | $y1$ $M\{nos\text{-}Cas9.P\}ZH\text{-}2A$ $w^*$;+; $y^{wing2+}$ $\Delta Nmnat/$ $TM6B$ | 607 | 16 | 5 | 21% |
| Nmnat:GFP: Nmnat $^{W129G}$ #1 | $y1$ $M\{nos\text{-}Cas9.P\}ZH\text{-}2A$ $w^*$;+; $y^{wing2+}$ $\Delta Nmnat/$ $TM6B$ | 653 | 0 | - | - |
| Nmnat:GFP: Nmnat $^{W129G}$ #2 | $y1$ $M\{nos\text{-}Cas9.P\}ZH\text{-}2A$ $w^*$;3 KB NMNAT GRC; $y^{wing2+}$ $\Delta Nmnat$ | 418 | 31 | 3 | 55% |
| Nmnat:GFP: Nmnat $^{\Delta251...257}$ | $y1$ $M\{nos\text{-}Cas9.P\}ZH\text{-}2A$ $w^*$;3 KB NMNAT GRC; $y^{wing2+}$ $\Delta Nmnat$ | 496 | 29 | 11 | 24% |
| Nmnat:GFP: Nmnat $^{C344S, C345S}$ | $y1$ $M\{nos\text{-}Cas9.P\}ZH\text{-}2A$ $w^*$;3 KB NMNAT GRC; $y^{wing2+}$ $\Delta Nmnat$ | 386 | 30 | 12 | 14% |
| Stub1:GFP | $y1$ $M\{nos\text{-}Cas9.P\}ZH\text{-}2A$ $w^*$; $y^{wing2+}$ $\Delta Stub1$ | 235 | 62 | 2 | 66% |
| CG11679:Flag | $y^{wing2+}$ $\Delta CG11679/FM7$ $Kgal4,UAS$ $GFP$;+/+; $attP2(y\text{-})\{nos\text{-}Cas9$ | 976 | 12* | 3 | 33% |
| Med27:flag | $y1$ $M\{nos\text{-}Cas9.P\}ZH\text{-}2A$ $w^*$;+; $y^{wing2+}$ $\Delta Med27$ | 833 | 17 | 3 | 29% |
| Amx:GFP | $y^{wing2+}$ $\Delta amx$;+/+; $attP2(y\text{-})\{nos\text{-}Cas9$ | 648 | 34 | 0 | - |

*excluding FM7 homozygotes and hemizygotes

DOI: https://doi.org/10.7554/eLife.38709.015

(*Zhai et al., 2006*). We then replaced the $y^{wing2+}$ SIC with internally GFP-tagged versions of wild-type and three variants of *Nmnat* (*Table 2*; *Figure 6A,B*). These variants of *Nmnat* are known or predicted to affect specific molecular functions of Nmnat *in vitro* (*Figure 6B*): (1) *Nmnat::GFP:: Nmnat^{W129G}* reduces NAD synthase activity (*Zhai et al., 2006*) (2) *Nmnat::GFP::Nmnat^{Δ251…257}* disrupts the ATP binding motif required for chaperone function (*Zhai et al., 2006*; *Ali et al., 2016*), and (3) *Nmnat::GFP::Nmnat^{C344S, C345S}* is predicted to disrupt critical palmitoylation sites that, in vertebrates, are required for membrane association and protein turnover (*Lau et al., 2010*; *Mayer et al., 2010*). All cassettes correctly replaced the $y^{wing2+}$ SIC based on Sanger sequencing.

We chose three independent lines of wild-type and each variant for further analysis. Homozygous *Nmnat::GFP::Nmnat^{WT}* flies are viable and fertile, suggesting that the internal GFP tag does not overtly affect Nmnat function. In contrast, homozygous *Nmnat::GFP::Nmnat^{Δ251…257}* or *Nmnat:: GFP::Nmnat^{Δ251…257}*/Nmnat^{Δ4790-1} animals die as 1^{st} instar larvae similar to *Nmnat^{ΔCDS,ywing2}*, suggesting that *Nmnat::GFP::Nmnat^{Δ251…257}* behaves as a null allele (*Zhai et al., 2006*). Homozygous *Nmnat::GFP::Nmnat^{W129G}* flies grow slowly and die prior to pupariation at late 3^{rd} instar larval stage, suggesting that it is a hypomorphic allele. Finally, the *Nmnat::GFP::Nmnat^{C344S, C345S}* flies that lack the putative palmitoylation sites are viable and fertile, but exhibit reduced lifespan relative to *Nmnat::GFP::Nmnat^{WT}* controls, indicating that it behaves as a weak hypomorph.

To determine protein levels and localization, we stained the brains of heterozygous GFP-tagged animals (*Figure 6C*). Antibody staining against GFP showed robust staining for wild-type *Nmnat:: GFP::Nmnat^{WT}* (*Figure 6C*). Most immunofluorescence signal is confined to the nucleus and cell body of neurons, but low levels of signal are observed in axons (*Figure 6C*). In contrast, *Nmnat:: GFP::Nmnat^{Δ251…257}* surprisingly produced no detectable protein (*Figure 6C*), suggesting that the removal of the ATP-binding domain severely affects the stability of the mRNA or protein, consistent with the observation that is a genetic null allele. On the other hand, *Nmnat::GFP::Nmnat^{W129G}* showed a mildly reduced signal when compared to wild-type, and the GFP localization was also seen mainly in the nucleus and cell body. Finally, *Nmnat::GFP::Nmnat^{C344S, C345S}* flies show an obvious increase in GFP levels consistent with the hypothesis that this site is required for protein degradation as previously shown in vertebrate NMNAT (*Mayer et al., 2010*; *Milde et al., 2013*; *Lau et al., 2010*). In summary, our data document the ability to perform structure-function analyses in the endogenously GFP tagged locus.

## Discussion

Here, we describe two methods to facilitate endogenous tagging and functional annotation of genes in *Drosophila*. DH doubles the success rate of RMCE using readily available MiMIC/CRIMIC lines to generate GFP protein traps or T2A-GAL4 gene traps. On the other hand, the $y^{wing2+}$ SIC-mediated two-step scarless gene tagging strategy offers a means to manipulate more than 6000 genes that cannot be targeted with the artificial exon approaches. Together these technologies facilitate the use of MiMICs and expand the capabilities of cassette swapping to include virtually all genes in *Drosophila*.

Recently, two other compound RMCE cassettes that encode two different modules in opposing orientations have been reported (*Fisher et al., 2017*; *Nagarkar-Jaiswal et al., 2017*). Flip-Flop contains a protein trap that can be inverted by Flp-FRT to a mutant allele, conditionally inactivating the gene in mitotic and postmitotic cells. The mutagenic module encodes *SA-T2A-mCherry-polyA* (*Nagarkar-Jaiswal et al., 2017*). In FlpStop on the other hand the non-mutagenic orientation does not encode a protein, but inversion by Flp leads to a gene trap *SA-stop cassette-polyA* (*Fisher et al., 2017*). Hence, these methods create conditional alleles. In contrast, DH creates alleles that are final and cannot be inverted or altered by Flp expression. This allows the use of Flp/FRT for independent manipulations in the DH background.

Previously, *Nagarkar-Jaiswal et al., 2015b* used a Flp-mediated *in vivo* mobilization strategy to create GFP-protein traps for intronic MiMIC containing genes whereas *Diao et al. (2015)* used a Cre-mediated *in vivo* mobilization scheme to create T2A-GAL4 gene traps. Both the GFP tag and T2A-GAL4 provide complementary means to assess gene function and expression. By combining the two in a single vector, DH greatly improves the rate of RMCE, the breadth of applications, and the amount of labor involved.

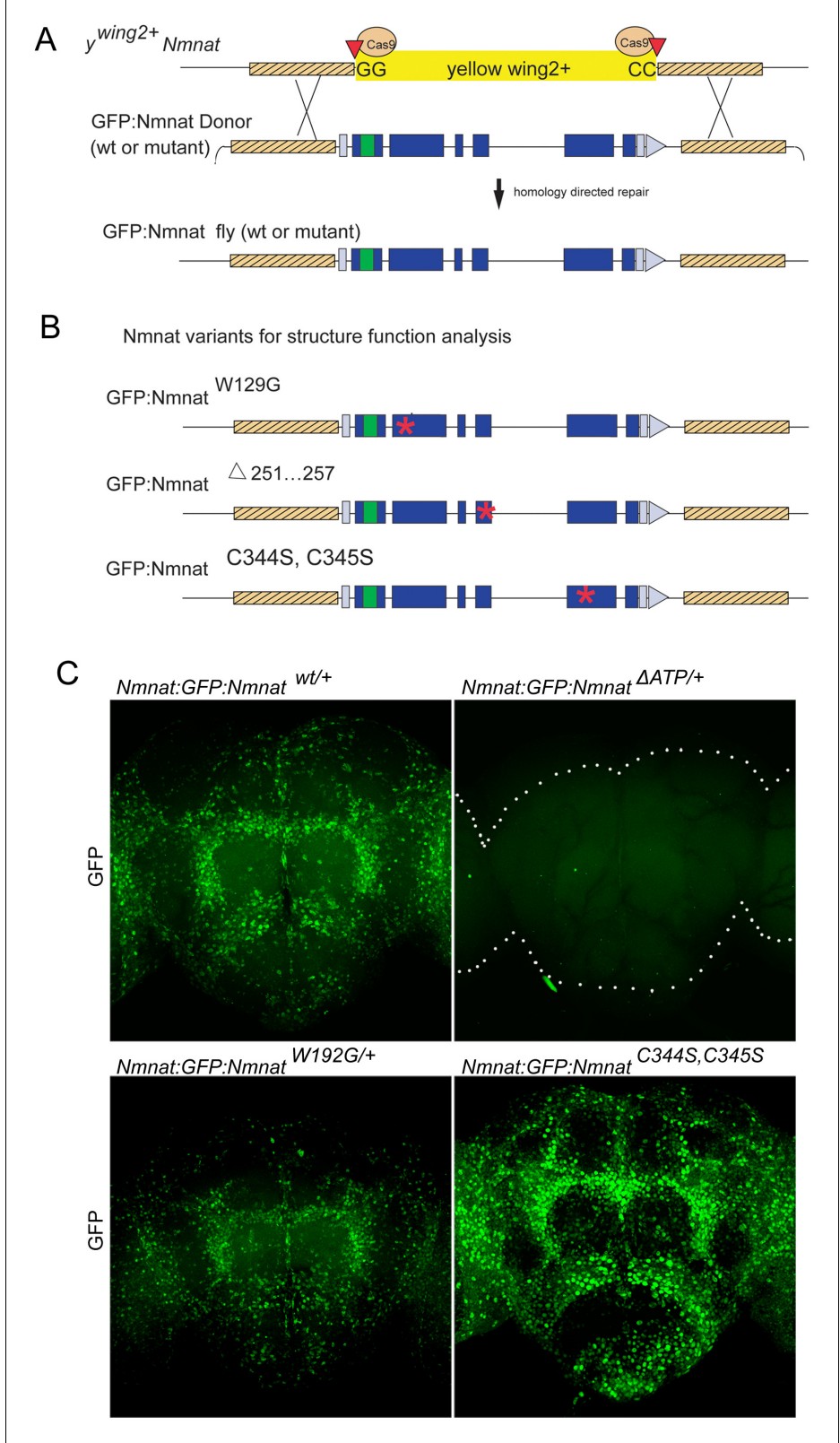

**Figure 6.** $y^{wing2+}$ cassette swapping facilitates structure-function analyses. (**A**) Schematic of the *Nmnat::GFP::Nmnat* donor construct for replacing the inserted $y^{wing2+}$ SIC at the *Nmnat* locus. (**B**) *Nmnat::GFP::Nmnat* variants used in the structure function experiment. Red * denotes approximate location of altered sequence(s). (**C**) Images
*Figure 6 continued on next page*

*Figure 6 continued*

of adult brains of *Nmnat::GFP::Nmnat* $^{wt}$ (Top left) *Nmnat::GFP::Nmnat* $^{W129G}$ (Bottom left) *Nmnat::GFP:: Nmnat* $^{\Delta251...257}$ (Top right) and *Nmnat:GFP:Nmnat* $^{C344S,\ C345S}$ (bottom right).

DOI: https://doi.org/10.7554/eLife.38709.016

We observed that although T2A-GAL4 is highly successful in marking the gene expression domains in the adult brain, for most genes the corresponding GFP-tagged protein signal in adult brains is often weak, consistent with previous results (*Lee et al., 2018a*; *Diao et al., 2015*). However, all the lines tested in the brain allow us to rapidly and reliably determine the cellular and subcellular localization of the GFP tagged proteins in egg chambers. Moreover, even when these GFP-tagged proteins cannot be used to detect the gene product, they can still be very useful for biochemical applications or to knock down the gene through a variety of methods to create conditional alleles (*Caussinus et al., 2011*; *Harmansa et al., 2017*; *Harmansa et al., 2015*; *Nagarkar-Jaiswal et al., 2015a*; *Neumüller et al., 2012*; *Lee et al., 2018b*).

Interestingly for three out of 28 MiMICs, we detected numerous DH RMCE inserts, judged by loss of the *yellow* marker, but for many of these events we could not determine conclusively the orientation or presence of DH by PCR. However, these false positives are easily identified by single fly PCR (*Figure 1—figure supplement 2*). Moreover, we could identify positive events in the MiMICs where the false positive rate was high, showing that the high false positive rate for these MiMICs does not limit the technique.

Finally, the GAL4 >UAS system can also be used to assess the function of cells, particularly neurons. Multiple features of neurons, including electrophysiological properties, can be modulated or assessed using established UAS constructs (*Venken et al., 2011b*). These include the UAS-Tetanus Toxin, UAS-Kir2.1 or UAS-Shibire$^{ts}$ to silence neurons (*Sweeney et al., 1995*; *Baines et al., 2001*; *Kitamoto, 2001*); UAS-TRPM8 or UAS-ChannelRhodopsin 2 (ChR2) to activate neurons (*Peabody et al., 2009*; *Schroll et al., 2006*); and UAS-GCaMP to assess changes in Calcium concentrations (*Chen et al., 2013*) or UAS-ASAP2 that acts as voltage sensor (*Yang et al., 2016*) to assess neuronal activity. Hence, the cells that express T2A-Gal4 associated with a specific gene can be manipulated in numerous ways.

Given that 50–60% of the protein coding genes do not contain suitable introns, numerous genes are not amenable for tagging based on our approaches. Inserting tags in genes that lack large (>150 bp) introns creates two main challenges: (1) screening for a precise rare gene editing event is very time consuming and (2) inserting extraneous sequences for RMCE within or near coding regions often create mutations and indels which disrupt protein function. Scarless gene editing offers obvious advantages for manipulating these loci, however, few options currently exist. Scarless gene editing can be achieved through the use of single-stranded DNA donors (*Gratz et al., 2014*; *Xue et al., 2014*). However, they are limited in size to ~200 nucleotides by current synthesis methods (*Korona et al., 2017*). Accordingly, sgRNA sites must be close to the specific nucleotides to be edited and because they cannot carry visible markers they require laborious screening methods to find flies carrying the correct gene editing event. Two strategies have been proposed to integrate fluorescent markers in fly genes using double stranded DNA donor plasmids and remove them to perform scarless genome editing (reviewed in *Bier et al., 2018*). One method, the Scarless-dsRed system (http://flycrispr.molbio.wisc.edu/scarless), relies on the *piggyBac* transposase to remove a dominant marker by precise excision after the gene editing event has been confirmed. However, as of yet no data have been reported to determine its efficiency or efficacy. While we were developing and testing our approach, a similar method, pGEM-wingGFP-tan, was reported (*Lamb et al., 2017*). This method integrates two Cas9 target sequences on either end of a GFP marker driven by a wing promotor to replace a locus. Unlike $y^{wing2+}$ which is readily visible in adults, the reported wing-GFP marker needs to be scored within a narrow developmental stage in pupae with a fluorescent microscope (*Lamb et al., 2017*). Moreover, *Lamb et al. (2017)* report a high rate of backbone insertion, where we observed only a single case out of 11 genes with our $y^{wing2+}$ approach. Since the methodology was only applied to a single gene, we cannot compare our data with *Lamb et al. (2017)*.

For 9 out of 10 genes that we targeted with $y^{wing2+}$, we obtained at least one correctly inserted SIC from an injection of ~500 embryos. The $y^{wing2+}$ marker is easy to score in adult flies and is

compatible with Golden Gate and Gibson assembly (*Engler et al., 2009*; *Gibson et al., 2009*), greatly facilitating its application to virtually any locus within the genome. The first step creates a null allele which provides an essential reference point for all subsequent genetic and molecular manipulations. Given that most genes that lack introns are rather small, they are poor targets for chemical or transposon mediated mutagenesis, although they can be targeted with CRISPR based on NHEJ. The $y^{wing2+}$ SIC offers an easy way to completely remove these small genes and is not labor intensive.

The major advantage of the $y^{wing2+}$ SIC is that it creates a highly versatile line. Multiple manipulations within the region of interest can be performed in parallel in an essentially isogenic background using this SIC. Although we did not observe widespread off-target mutations, we suggest the use of multiple lines where possible. We have shown that the cassette swapping via the $y^{wing2+}$ SIC occurs precisely with a high success rate and demonstrated its usefulness both for gene tagging and for structure function analysis.

In summary, the two methodologies and accompanying tool kits presented here complement and expand existing MiMIC and CRIMIC approaches. The combination of these methodologies should enable endogenous tagging and manipulation of most fly genes, an invaluable resource for the fly research community.

# Materials and methods

**Key resources table**

| Reagent type (species) or resource | Designation | Source or reference | Identifiers | Additional information |
|---|---|---|---|---|
| Genetic reagent (*Drosophila melanogaster*) | Double Header Jump Starter phase 0 on chromosome II | This study | | Fly strain containing DH flanked by LoxP sites |
| Genetic reagent (*D. melanogaster*) | Double Header Jump Starter phase one on chromosome II | This study | | Fly strain containing DH flanked by LoxP sites |
| Genetic reagent (*D. melanogaster*) | Double Header Jump Starter phase two on chromosome II | This study | | Fly strain containing DH flanked by LoxP sites |
| Genetic reagent (*D. melanogaster*) | Double Header Jump Starter phase 0 on chromosome III | This study | | Fly strain containing DH flanked by LoxP sites |
| Genetic reagent (*D. melanogaster*) | Double Header Jump Starter phase one on chromosome III | This study | | Fly strain containing DH flanked by LoxP sites |
| Genetic reagent (*D. melanogaster*) | Double Header Jump Starter phase two on chromosome III | This study | | Fly strain containing DH flanked by LoxP sites |
| Genetic reagent (*D. melanogaster*) | MI01487 (kibra) | *Venken et al. (2011a)*; *Nagarkar-Jaiswal et al., 2015a* | flybaseID#_FBst0040175; RRID:BDSC_40175 | |
| Genetic reagent (*D. melanogaster*) | MI05208 [5-HT2B (5-hydroxytryptamine receptor 2B)] | *Venken et al. (2011a)*; *Nagarkar-Jaiswal et al., 2015a* | flybaseID#_FBst0042994; RRID:BDSC_42994 | |
| Genetic reagent (*D. melanogaster*) | MI06794 [Lgr4(Leucine-rich repeat-containing G protein-coupled receptor 4)] | *Venken et al. (2011a)*; *Nagarkar-Jaiswal et al., 2015a* | flybaseID#_FBst0042179: RRID:BDSC_42179 | |
| Genetic reagent (*D. melanogaster*) | MI06872 (CG34383) | *Venken et al. (2011a)*; *Nagarkar-Jaiswal et al., 2015a* | flybaseID#_FBst0041111; RRID:BDSC_41111 | |
| Genetic reagent (*D. melanogaster*) | MI08614 [Dgk (Diacyl glycerol kinase)] | *Venken et al. (2011a)*; *Nagarkar-Jaiswal et al., 2015a* | flybaseID#_FBst0044991; RRID:BDSC_44991 | |

*Continued on next page*

*Continued*

| Reagent type (species) or resource | Designation | Source or reference | Identifiers | Additional information |
|---|---|---|---|---|
| Genetic reagent (*D. melanogaster*) | MI11741 (CG12206) | *Venken et al. (2011a); Nagarkar-Jaiswal et al., 2015a* | flybaseID#_FBst0056687; RRID:BDSC_56687 | |
| Genetic reagent (*D. melanogaster*) | MI15073 (CG9132) | *Venken et al. (2011a); Nagarkar-Jaiswal et al., 2015a* | flybaseID#_FBst0059739; RRID:BDSC_59739 | |
| Genetic reagent (*D. melanogaster*) | MI02926 (Pits) | *Venken et al. (2011a); Nagarkar-Jaiswal et al., 2015a* | flybaseID#_FBst0036165; RRID:BDSC_36165 | |
| Genetic reagent (*D. melanogaster*) | MI00805 (CG6966) | *Venken et al. (2011a); Nagarkar-Jaiswal et al., 2015a* | flybaseID#_FBst0034113; RRID:BDSC_34113 | |
| Genetic reagent (*D. melanogaster*) | MI12643 (fz) | *Venken et al. (2011a); Nagarkar-Jaiswal et al., 2015a* | flybaseID#_FBst0058645; RRID:BDSC_58645 | |
| Genetic reagent (*D. melanogaster*) | MI05871 (Doa) | *Venken et al. (2011a); Nagarkar-Jaiswal et al., 2015a* | flybaseID#_FBst0043880; RRID:BDSC_43880 | |
| Genetic reagent (*D. melanogaster*) | MI08818 (qless) | *Venken et al. (2011a); Nagarkar-Jaiswal et al., 2015a* | flybaseID#_FBst0051110; RRID:BDSC_51110 | |
| Genetic reagent (*D. melanogaster*) | MI06179 (DCX-EMAP) | *Venken et al. (2011a); Nagarkar-Jaiswal et al., 2015a* | flybaseID#_FBst0043047; RRID:BDSC_43047 | |
| Genetic reagent (*D. melanogaster*) | MI14396 (CG6293) | *Venken et al. (2011a); Nagarkar-Jaiswal et al., 2015a* | flybaseID#_FBst0059511; RRID:BDSC_59511 | |
| Genetic reagent (*D. melanogaster*) | MI00445 (Nlg3) | *Venken et al. (2011a); Nagarkar-Jaiswal et al., 2015a* | flybaseID#_FBst0031005; RRID:BDSC_31005 | |
| Genetic reagent (*D. melanogaster*) | MI09222 (CG1578) | *Venken et al. (2011a); Nagarkar-Jaiswal et al., 2015a* | flybaseID#_FBst0051263; RRID:BDSC_51263 | |
| Genetic reagent (*D. melanogaster*) | MI00494 (wnd) | *Venken et al. (2011a); Nagarkar-Jaiswal et al., 2015b* | flybaseID#_FBst0031028; RRID:BDSC_31028 | |
| Genetic reagent (*D. melanogaster*) | MI02915 (Ask1) | *Venken et al. (2011a); Nagarkar-Jaiswal et al., 2015a* | flybaseID#_FBst0036163; RRID:BDSC_36163 | |
| Genetic reagent (*D. melanogaster*) | MI03136 (LPCAT) | *Venken et al. (2011a); Nagarkar-Jaiswal et al., 2015a* | flybaseID#_FBst0036425; RRID:BDSC_36425 | |
| Gnetic reagent (*D. melanogaster*) | MI10071 (Trpl) | *Venken et al. (2011a); Nagarkar-Jaiswal et al., 2015a* | flybaseID#_FBst0053455; RRID:BDSC_53455 | |
| Genetic reagent (*D. melanogaster*) | MI09900 (Sap47) | *Venken et al. (2011a); Nagarkar-Jaiswal et al., 2015a* | flybaseID#_FBst0053794; RRID:BDSC_53794 | |
| Genetic reagent (*D. melanogaster*) | MI01646 (CG1815) | *Venken et al. (2011a); Nagarkar-Jaiswal et al., 2015a* | flybaseID#_FBst0035948; RRID:BDSC_35948 | |
| Genetic reagent (*D. melanogaster*) | MI04010 (Tbh) | *Venken et al. (2011a); Nagarkar-Jaiswal et al., 2015a* | flybaseID#_FBst0056660; RRID:BDSC_56660 | |

*Continued on next page*

*Continued*

| Reagent type (species) or resource | Designation | Source or reference | Identifiers | Additional information |
|---|---|---|---|---|
| Genetic reagent (*D. melanogaster*) | MI13728 (CG17841) | *Venken et al. (2011a)*; *Nagarkar-Jaiswal et al., 2015a* | flybaseID#_ FBst0059189; RRID:BDSC_59189 | |
| Genetic reagent (*D. melanogaster*) | MI05741 (CG1632) | *Venken et al. (2011a)*; *Nagarkar-Jaiswal et al., 2015a* | flybaseID#_ FBst0042106; RRID:BDSC_42106 | |
| Genetic reagent (*D. melanogaster*) | MI10889 (CG17167) | *Venken et al. (2011a)*; *Nagarkar-Jaiswal et al., 2015a* | flybaseID#_ FBst0056092; RRID:BDSC_56092 | |
| Genetic reagent (*D. melanogaster*) | MI00986 (CG32698) | *Venken et al. (2011a)*; *Nagarkar-Jaiswal et al., 2015a* | flybaseID#_ FBst0035095; RRID:BDSC_35095 | |
| Genetic reagent (*D. melanogaster*) | MI15214 (CG13375) | *Venken et al. (2011a)*; *Nagarkar-Jaiswal et al., 2015a* | flybaseID#_ FBst0060995; RRID:BDSC_60995 | |
| Genetic reagent (*D. melanogaster*) | Nmnat$^{ywing2+}$ | This study | | fly strain carrying the *ywing2+* dominant marker replacing the gene*Nmnat* |
| Genetic reagent (*D. melanogaster*) | Stub1$^{ywing2+}$ | This study | | fly strain carrying the *ywing2+* dominant marker replacing the gene Stub1 |
| Genetic reagent (*D. melanogaster*) | Ubqn$^{ywing2+}$ | This study | | fly strain carrying the *ywing2+* dominant marker replacing the gene Ubqn |
| Genetic reagent (*D. melanogaster*) | Itp-r83A$^{ywing2+}$ | This study | | fly strain carrying the *ywing2+* dominant marker replacing the gene Itp-r83 |
| Genetic reagent (*D. melanogaster*) | CG18769$^{ywing2+}$ | This study | | fly strain carrying the *ywing2+* dominant marker replacing the gene CG18769 |
| Genetic reagent (*D. melanogaster*) | CG13390$^{ywing2+}$ | This study | | fly strain carrying the *ywing2+* dominant marker replacing the gene CG13390 |
| Genetic reagent (*D. melanogaster*) | Med27$^{ywing2+}$ | This study | | fly strain carrying the *ywing2+* dominant marker replacing the gene Med27 |
| Genetic reagent (*D. melanogaster*) | CG11679$^{ywing2+}$ | This study | | fly strain carrying the *ywing2+* dominant marker replacing the gene CG11679 |
| Genetic reagent (*D. melanogaster*) | amx$^{ywing2+}$ | This study | | fly strain carrying the *ywing2+* dominant marker replacing the gene amx |
| Genetic reagent (*D. melanogaster*) | *Nmnat::GFP::Nmnat$^{WT}$* | This study | | fly strain carrying the Nmnat gene with S(GSS)$_4$... EGFP coding sequence... (GSS)$_4$ integrated internally into the protein between 3R:24945353 and 3R:24945353 |
| Genetic reagent (*D. melanogaster*) | *Nmnat::GFP::Nmnat$^{W129G}$* | This study | | fly strain carrying the Nmnat gene with S(GSS)$_4$...EGFP coding sequence...(GSS)$_4$ integrated internally into the protein between 3R: 24945353 and 3R:24945353 and bearing a mutation producing W192G |

*Continued on next page*

Genetics and Genomics

*Continued*

| Reagent type (species) or resource | Designation | Source or reference | Identifiers | Additional information |
|---|---|---|---|---|
| Genetic reagent (*D. melanogaster*) | *Nmnat::GFP::Nmnat*$^{\Delta251\ldots257}$ | This study | | fly strain carrying the Nmnat gene with S(GSS)$_4$...EGFP coding sequence…(GSS)$_4$ integrated internally into the protein between 3R: 24945353 and 3R:24945353 and bearing a deletion removing amino acids 251…257 |
| Genetic reagent (*D. melanogaster*) | *Nmnat::GFP::Nmnat*$^{C344S,\ C345S}$ | This study | | fly strain carrying the Nmnat gene with S(GSS)$_4$... EGFP coding sequence… (GSS)$_4$ integrated internally into the protein between 3R: 24945353 and 3R:24945353 and bearing a mutation producing C344S, C345S |
| Recombinant DNA reagent | Double Header | This study | | Recombination Mediated Cassette Exchange donor plasmid containing SA-T2A -GAL4-polyA and SA-GFP-SD in the opposite orientation |
| Recombinant DNA reagent | pattB *y*$^{wing2+}$ | This study | | Vector for φC31 integrated transgenesis that expresses the *yellow* gene product in the wings |
| Recombinant DNA reagent | pattB *y*$^{body+}$ | This study | | Vector for φC31 integrated transgenesis that expresses the *yellow* gene product in the body |
| Recombinant DNA reagent | p{*y*$^{wing2+}$} | This study | | Donor vector compatible with Golden Gate cloning carrying the *yellow* wing2 + dominant marker flanked by nucleotides 'GG' and 'CC' upstream and downstream, respectively |
| Recombinant DNA reagent | p{*y*$^{body+}$} | This study | | Donor vector compatible with Golden Gate cloning carrying the *yellow* body dominant marker flanked by nucleotides 'GG' and 'CC' upstream and downstream, respectively |
| Recombinant DNA reagent | p{EGFP Donor} | This study | | Donor vector compatible with Golden Gate cloning carrying the EGFP coding sequence flanked by (GSS) linker sequences |
| Recombinant DNA reagent | p{mCherry Donor} | This study | | Donor vector compatible with Golden Gate cloning carrying the mCherry coding sequence flanked by (GSS) linker sequences |
| Recombinant DNA reagent | p{T2a-GAL4 Donor}} | This study | | Donor vector compatible with Golden Gate cloning carrying the T2a viral peptide sequence in frame with the GAL4 transcription factor coding sequence |

*Continued on next page*

*Continued*

| Reagent type (species) or resource | Designation | Source or reference | Identifiers | Additional information |
|---|---|---|---|---|
| Recombinant DNA reagent | p{T2a-GAL4-PolyA Donor}} | This study | | Donor vector compatible with Golden Gate cloning carrying the T2a viral peptide sequence in frame with the GAL4 transcription factor coding sequence terminating in the SV40 transcriptional terminator |
| Recombinant DNA reagent | pCFD3-dU6:3gRNA | Port et al. (2014) | Addgene_plasmid_#49410 | |
| Recombinant DNA reagent | Double Header | This study | | Recombination Mediated Cassette Exchange donor plasmid containing SA-T2A-GAL4-polyA and SA-GFP-SD in the opposite orientation |
| Antibody | anti-GFP antibody conjugated with FITC | Abcam | RRID: AB_305635 | used at 1:500 |
| Antibody | anti-GFP | Invitrogen | Cat#_ A11122 | used at 1:500 |

## Cloning of DH

Sequence of the DH plasmids can be found in Supp. file.

Briefly, SA- EGFP-FlAsH-StrepII-TEVcs-3xFlag-SD cassette was PCR amplified from pBS-KS-attB1-2-PT-SA-SD-pX (corresponding to the codon phase)-EGFP-FlAsH-StrepII-TEV-3xFlag (DGRC # 1298, [*Venken et al., 2011b*]) with tags_for_BsiWI, tags_rev_AvrII primers. This fragment is cloned in, a modified pTGEM plasmid of pX (corresponding to the codon phase) where the loxP site before 3XP3RFP is deleted and a BsiWI site is integrated after an AvrII site using BsiWI and AvrII restriction sites. Resulting vector is cut with XbaI-BsiWI and cloned into pC-(loxP2-attB2-SA(1)-T2A-Gal4-Hsp70) (Addgene # 62955, [*Diao et al., 2015*]) modified to include a BsiWI site after SD, generating DH pX.

## Cloning of constructs to test *yellow* wing enhancer expression

A sub-region of the *yellow* dominant marker from P{EPgy2} (*Bellen et al., 2011*) that contains the promoter, coding sequence and UTRs was subcloned into the plasmid pattB (Accession # KC896839 [*Bischof et al., 2013*]) using oligos DLK0048 and DLK0049 (see *Supplementary file 1* for table showing oligonucleotides sequences used) flanked by XhoI and XbaI to make the vector pBS II SK-attB yMP *w*+. Either the full sequence of the *yellow* enhancers (oligos DLK0054 and DLK0056) (*Geyer and Corces, 1987*) or sub-fragments (oligos DLK0054 and DLK0057 or DLK0055 and DLK0056 - see *Figure 5—figure supplement 2*) were then subcloned into pattB yMP *w*+ to make pattB expression constructs. Once expressions of the markers were verified, *miniwhite*+ and the *loxP* sites were removed by digestion with ApaI and NotI, the ends blunted with DNA Polymerase I, Large (Klenow) Fragment (NEB) and ligated using T4 Ligase (NEB). As only the full body, *full wing* and *wing2* gave positive expression, *wing2* was chosen for use as the most compact construct and a multiple cloning site was then added using annealed oligos DLK0022 and DLK0023 cut with SpeI and XbaI into XbaI digested pattB $y^{wing2+}$, and sequence verified for insert direction to make functional plasmids for $\phi$C31 mediated transgenesis. The sequence of pattB $y^{wing2+}$ can be found in *Supplementary file 1*.

## Cloning $y^{wing2+}$ Golden Gate donor template

Three versions of p{$y^{wing2+}$} were cloned: for BsmBI, BsaI, and BbsI. First, annealed oligos (DLK320 and DLK338, DLK322 and DLK339, and DLK324 and DLK340 for BsmBI, BbsI and BsaI, respectively) containing the appropriate SacI overhang and ends-in TypeIIS restriction sites with a short intervening random nucleotide spacer inserted into the pM14 plasmid backbone (*Lee et al., 2018a*) digested with enzymes SacI and EcoRV. The $y^{wing2+}$ dominant reporter was then subcloned by PCR

using oligos DLK326 and DLK327 from p-attB $y^{wing2+}$ into the pM14 + spacer backbone digested with BbsI, BsaI or BsmBI. The sequence of p{$y^{wing2+}$} can be found in *Supplementary file 1*.

### Cloning the GFP, mCherry and T2A-GAL4 donor templates

Three versions of p{*GFP*} were cloned: for BsmBI, BsaI, and BbsI. Annealed oligos (DLK461 and DLK462, DLK463 and DLK464, and DLK465 and DLK466 for BsmBI, BbsI and BsaI, respectively) containing the appropriate SacI overhang and ends-in TypeIIS restriction sites were inserted into the pM14 plasmid backbone (*Lee et al., 2018a*) digested with enzymes SacI and EcoRV to make p{spacer-L-L} where L denotes (GGS)$_4$ linker. The linker-GFP sequence was generated by PCR from pM14 (*Lee et al., 2018b*) and using oligos DLK225 and DLK300 and subcloned into p{spacer-L-GFP-L}. The second linker was then added from annealed oligos DLK554 and DLK555 to make the BbsI version of p{spacerL–L}. Finally, the linker-GFP-linker was subcloned into versions of p{spacer-L -L} for BsaI and BsmBI to make plasmids compatible with each enzyme. Additional template vectors were produced for mCherry and T2A-GAL4 inserts which can also be found in Supp. file. mCherry was subcloned into p{spacer-L-L} from the Flip-Flop cassette which was designed with silent mutations to remove BsaI sites (*Nagarkar-Jaiswal et al., 2017*) and T2A-GAL4 inserts were subcloned from pM14 (*Lee et al., 2018b*) into p{spacer-T2A}. Sequences can be found in *Supplementary file 1*.

### Cloning HDR donor injection plasmids

Donor constructs were generated as previously described (*Housden and Perrimon, 2016*). Briefly, homology arms were PCR amplified from genomic DNA using Q5 polymerase (NEB), run on an agarose gel and purified with the QIAquick Gel Extraction Kit (Qiagen). The homology arms, pBH donor vector and either p{$y^{wing2}$} or p{L-GFP-L} cassette were combined by Golden Gate assembly (*Engler et al., 2009*) using the appropriate type IIS restriction enzyme (BbsI, BsaI, or BsmBI). The resulting reaction products were transformed into Stbl2 Chemically Competent Cells (ThermoFisher), and plated overnight under kanamycin selection. Colonies were cultured for 24 hr at 30°C and DNA was prepared by miniprep (QIAGEN). The entire homology arm and partial sequences of the adjacent cassette were verified prior to injection. Additional cloning information and sequences for all HDR donor plasmids can be found in Supp. file.

### Cloning sgRNA expression constructs

sgRNA expression constructs were cloned into the vector pCFD3-dU6:3gRNA (Addgene plasmid #49410, *Port et al., 2014*) using established protocols (http://www.crisprflydesign.org/wp-content/uploads/2014/05/Cloning-with-pCFD3.pdf). Sequences of sgRNAs can be found in *Supplementary file 1*.

### Cloning genomic rescue constructs

Genomic fragments were cloned into pattB (Ascension # KC896839 (*Bischof et al., 2013*) by PCR from wild-type genomic DNA and inserted BamHI/NotI for CG13390 (oligos DLK823 and DLK824) and XhoI/AvrII (into the XbaI site of pattB) for Med27 (oligos DLK966 and DLK969)

### Fly lines

MiMIC stocks are obtained from Bloomington Drosophila Stock Center.

The fly lines to conduct RMCE of DH with crosses are:

| |
|---|
| *y[1] M{vas-int.Dm}ZH-2A w[*], P{y[+mDint2]=Crey}1b;; Sb[1]/TM2, Ubx[130] e[s]* |
| *y[1] M{vas-int.Dm}ZH-2A w[*], P{y[+mDint2]=Crey}1b; sna[Sco]/CyO* |
| *y[1] w[*]; P{DHp0-7 [w+]}; TM2 Ubx[130] e[s]/TM6, Tb[1] e[s]* |
| *y[1] w[*]; P{DHp1-7 [w+]}; TM2 Ubx[130] e[s]/TM6, Tb[1] e[s]* |
| *y[1] w[*]; P{DHp2-4 [w+]}; TM2 Ubx[130] e[s]/TM6, Tb[1] e[s]* |
| *y[1] w[*]; Kr[If1] wg[Sp1]/CyO; P{DHp0-8 [w+]}* |
| *y[1] w[*]; Kr[If1] wg[Sp1]/CyO; P{DHp1-6 [w+]}* |
| *y[1] w[*]; Kr[If1] wg[Sp1]/CyO; P{DHp2-6A [w+]}* |

Cas9 stocks for CRISPR experiments carried an isogenic chromosome on either X, II or III derived from $y\ w$ and were either $y^1$ $M\{nos\text{-}Cas9.P\}ZH\text{-}2A\ w^*$ (from Bloomington Drosophila Stock Center) or $y^1$ (iso X) $w^*$; +/+; attP2(y+){nos-Cas9(y+)} (*Ren et al., 2013*) in which the $y$ + marker was mutagenized by injecting sgRNA expression plasmids (pCFD3-y1 and pCFD3-y2) against the *yellow* coding sequence. The isogenic chromosomes (X, II, III) were sequenced using whole genome sequencing (Human Genome Sequencing Center at Baylor College of Medicine). The sequence (.BAM) files are available on Zenodo (https://zenodo.org/record/1341241)."

## Generation of Double Header transgenics

RMCE to generate DH insertions by injections is depicted in *Figure 1—figure supplement 1A* and in (*Nagarkar-Jaiswal et al., 2017*). Briefly, the DH plasmid of the correct phase (500 ng/ul final concentration) is mixed with ΦC31 integrase helper plasmid (400 ng/ul final concentration) and injected in embryos of MiMIC stocks. The crossing scheme for generating DH insertions is depicted in *Figure 2—figure supplement 1*. 5–7 crosses with 10–15 virgins of individual MiMIC lines are crossed with 7–10 males from phase appropriate DH transgenics on second chromosome, balanced for third chromosome. The vials are flipped every second day to prevent overcrowding. 5–10 crosses are set in the subsequent generations. The resulting individual $y$- flies are selected to set up stocks. Note that SICs are flanked by FRT sites in the newly generated CRIMIC alleles. Hence, the Flp/FRT mobilization schemes described by *Nagarkar-Jaiswal et al., 2015b* cannot be used to generate GFP protein traps in these CRIMICs, given that expression of the Flp would excise the SIC. For the same reason, Flp/FRT cannot be used in combination with the CRIMIC T2A-GAL4 alleles for experiments that require both T2A-GAL4 and Flp.

## PCR determination of DH orientation

For each DH insertion stock four PCRs are set. Primer pairs are MiMIC_5'_for-GFP_DH_for, MiMIC_3'_rev-T2A_GAL4_rev, MiMIC_5'_for-T2A_GAL4_rev, and MiMIC_3'_rev-GFP_DH_for. Correct RMCE events result in 2 out of 4 successful amplicons (*Figure 1—figure supplement 2*). We isolated a number of lines where only one out of two amplicons was detected or gave conflicting results. These lines are not included in the analysis in this manuscript.

## CRISPR/Cas9 injections for scarless $y^{wing2+}$ gene editing

Embryos (designated $G_0$) at less than one hour post-egg laying carrying the appropriate Cas9 allele were injected with a mixture of 150–200 ng/μl of donor plasmid and 25 ng/μl of each sgRNA expression construct, transferred to standard media after 24–48 hr, and crossed to $y\ w$ flies. For experiments to insert the $y^{wing2+}$ cassette, offspring were screened for presence of $yellow^+$ wings several days after eclosion and crossed to appropriate balancers. For experiments to replace the $y^{wing2+}$ cassette, $G_0$ flies were crossed to $y\ w$ flies carrying appropriate balancers and F1 offspring screened for loss of $yellow+$ wings. Individual F1 founders were backcrossed to $y\ w$ flies carrying appropriate balancers, allowed to establish larvae, and screened for presence of the insert via PCR (see *Supplementary file 1*).

## Confocal imaging

Confocal imaging was conducted as in the previous study (*Lee et al., 2018a*). In brief, dissected adult brains were fixed in 4% paraformaldehyde/1xPBS overnight, then penetrated with 0.2% Triton X-100/1xPBS at 4°C overnight. The larval brains or other tissues were fixed in 4% paraformaldehyde/1 xPBS at 4°C for at least 2 hr, transferred to 0.5% Triton X-100/1xPBS for overnight 4°C incubation. For adult brains, the samples were vacuumed for 1 hr at room temperature, and left overnight in the same solution for penetration at 4°C. For immunostaining of GFP, the samples were incubated with anti-GFP antibody conjugated with FITC (1:500) (Abcam, RRID: AB_305635) in 1xPBS with 0.5% Triton X-100 overnight. To increase signal, some samples used anti-GFP antibody (1:1000) (Invitrogen, A11122) followed by incubation with secondary antibody (Alexa Fluor 488-conjugated goat anti-rabbit IgG). Samples were cleared and mounted in RapiClear (SunJin Lab Co.) and imaged with a Zeiss LSM 880 under a Plan-Apochromat 20x/0.8 M27 objective with numerical aperture of 0.8 or Leica Sp8 Confocal Microscope under a HC PL APO 20x objective with numerical aperture 0.7. Laser

intensity and detector gains were adjusted as needed to increase signal-to-noise ratio and prevent signal saturation.

## Acknowledgements

We thank the Bloomington *Drosophila* Stock Center and the Developmental Studies Hybridoma Bank for antibodies. We thank Zelun Wang for technical help, and Megan Campbell and Shinya Yamamoto for reading the manuscript and providing helpful suggestions as well as Claire Hu for valuable information. Confocal microscopy was performed in the BCM IDDRC Neurovisualization Core, supported by the NICHD (U54HD083092). This research was supported by NIH grant R01GM067858, The Robert A and Renee E Belfer Family Foundation, and the Huffington Foundation to HJB. DLK was supported by the Brain Disorders training grant NIH T32NS043124. HJB is an investigator of the Howard Hughes Medical Institute.

## Additional information

### Competing interests

Hugo J Bellen: Reviewing editor, *eLife*. The other authors declare that no competing interests exist.

### Funding

| Funder | Grant reference number | Author |
| --- | --- | --- |
| National Institutes of Health | R01GM067858 | Hugo J Bellen |
| Howard Hughes Medical Institute | | Hugo J Bellen |
| Robert A. and Renee E. Belfer Family Foundation | | Hugo J Bellen |
| Huffington Foundation | | Hugo J Bellen |
| National Institutes of Health | Training grant T32NS043124 | David Li-Kroeger |

The funders had no role in study design, data collection and interpretation, or the decision to submit the work for publication.

### Author contributions

David Li-Kroeger, Oguz Kanca, Conceptualization, Data curation, Investigation, Visualization, Methodology, Writing—original draft, Writing—review and editing; Pei-Tseng Lee, Sierra Cowan, Michael T Lee, Manish Jaiswal, Jose Luis Salazar, Yuchun He, Zhongyuan Zuo, Investigation, Writing—review and editing; Hugo J Bellen, Resources, Supervision, Funding acquisition, Project administration, Writing—review and editing

### Author ORCIDs

David Li-Kroeger http://orcid.org/0000-0001-6473-7691
Pei-Tseng Lee http://orcid.org/0000-0002-7501-7881
Sierra Cowan http://orcid.org/0000-0003-3530-9326
Hugo J Bellen http://orcid.org/0000-0001-5992-5989

### Decision letter and Author response

Decision letter https://doi.org/10.7554/eLife.38709.022
Author response https://doi.org/10.7554/eLife.38709.023

## Additional files

### Supplementary files

• Supplementary file 1. Sequences of oligonucleotides, sgRNAs and key vectors used in this study are shown along with a protocol for designing donor templates using the *yellow* wing2+ swappable insertion cassette.

DOI: https://doi.org/10.7554/eLife.38709.017

• Transparent reporting form

DOI: https://doi.org/10.7554/eLife.38709.018

### Data availability

All data generated or analyzed during this study are included in the manuscript and supporting files. The whole genome sequencing files are available on Zenodo (https://zenodo.org/record/1341241).

The following dataset was generated:

| Author(s) | Year | Dataset title | Dataset URL | Database, license, and accessibility information |
|---|---|---|---|---|
| Li-Kroeger D, Kanca O, Lee P-T, Cowan S, Lee M, Jaiswal M, Salazar JL, He Y, Zuo Z, Bellen HJ | 2018 | An expanded toolkit for gene tagging based on MiMIC and scarless CRISPR tagging in Drosophila - sequence files | https://zenodo.org/record/1341241 | Publicly available on Zenodo under a Creative Commons Attribution 4.0 License |

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
