## [Decision Letter]

Thank you for submitting your article "An expanded toolkit for gene tagging based on MiMIC and scarless CRISPR tagging in *Drosophila*" for consideration by *eLife*. Your article has been reviewed by three peer reviewers, including Yukiko M Yamashita as the Reviewing Editor and Reviewer #1, and the evaluation has been overseen by K VijayRaghavan as the Senior Editor.

The reviewers have discussed the reviews with one another and the Reviewing Editor has drafted this decision to help you prepare a revised submission.

Summary:

This is yet another set of useful tools by the Bellen lab allowing the fly community to move on and fully profit from the new CRISPR technologies in full, and the reviewers are quite positive about this work. Reviewer #3 pointed out that the image quality can be better, and if it is something that can be easily addressed, better images should be used. We recognize that it is possible that the authors already tried to optimize imaging quality but the Materials and methods section does not provide sufficient information whether the image can be further improved or not. Thus, we would like to see 1) better images and 2) more information about the imaging in the Materials and methods section.

Reviewer #1:

This is a report on yet another useful gene trap/protein trap methodology that utilizes MiMIC in *Drosophila*.

This new resource is named 'Double Header' (DH), which results in protein trap OR gene trap (gal4) depending on which direction insertion happens, making any insertion events 'productive'. DH construct can be provided either as a plasmid to be injected to the fly embryos, or as Cre-mediated circularized DNA from the genomic transgene. The latter only requires genetic crosses. They provide a few 'proof of principle' examples and confirm that DH behaves as expected (e.g. reporting the expression pattern). DH and characterized a few DH insertions on known genes. Then they turned their attention to a method that can target genes that do not contain introns at all (which are about ~50% of *Drosophila* genes, but can never be targeted by MiMIC strategy). They used CRISPR/Cas9 strategy to replace a gene of interest with visible, dominant marker (they chose y gene expressed in wing) to facilitate visual screening. Then this will be next used to be replaced with any donor gene (as an example, they generated point mutations in Nmnat gene).

There is no doubt that DH is another extremely useful reagent developed by Bellen lab, and I do not have further comments. But it was not too clear to me whether the second method (two-step replacement of the gene) is significantly faster or more convenient than other already-applied CRISPR methods to generate endogenous tag. This method involves 1) generation of a gene specific construct to generate knock-out flies, and 2) generation of a construct that can tag or point-mutate the gene of interest, therefore need to be conducted at gene-by-gene basis. But I do see a point that DH and CRISPR methods are two complementary methods that are intended to cover all protein coding genes in the *Drosophila* genome, so it makes sense to report in a single paper. They have already done their best to explain the advantages of their CRISPR method, so I don't have additional suggestions, but I am just noting that I was a bit puzzled when they moved onto CRISPR method because two methods are very different.

Reviewer #2:

This is yet another set of useful tools by the Bellen lab allowing the fly community to move on and fully profit from the new CRISPR technologies in full. As for all the other recently developed tools and resources, these novel methods will be used by many to perform precise gene manipulation and to facilitate the monitoring of gene expression, protein localisation, as well as providing the reagents for many other protein-based manipulations. I do not see any need for further work or editing.

Reviewer #3:

Li-Kroeger and colleagues describe two very useful extensions to widely used current approaches to genome engineering of individual *Drosophila* genes.

The first one is an incremental (but still productive) improvement to the widely used MiMIC system. The authors design a single insertional fragment that can insert either as an in-frame fusion (in any reading frame) or a GAL4 transcriptional fusion, in a single experiment, and show that this can work reasonably efficiently when injected, and more efficiently when supplied from an existing transgenic construct. While the advance is an incremental one rather than a new concept, it is likely to be widely used, since it can potentially generate pairs of very useful stocks, for the same effort currently used to generate either a single GFP or GAL4 insertion, it can be applied using thousands of existing MiMIC insertion stocks that are already available from stock centres, and only a few additional transgenic stocks described in this paper – which the authors have a good track record of sharing with the community.

The second reported advance is more substantial, both conceptually and practically. It provides a CRISPR-based approach to overcome some of the limitations of the MiMIC system and other CRISPR approaches. They describe a two-step process, in which CRISPR is first used to integrate a yellow+ cassette at a gene of interest, and the then locus with the inserted cassette is again targeted by CRISPR to integrate an engineered fusion and/or mutant allele of interest. The advantages are:

- use of a convenient body color marker to score each of the two steps, and identify low-frequency events more conveniently than by molecular screening;

- the ability to target any gene, independent of whether existing insertional mutations are available;

- design of the insertional cassette sequences which makes the insertion or swap steps irreversible;

- the ease of introducing any mutations or fusions constructed in vitro;

- extensive supporting evidence from several genes, and a different mutations or fusions with a gene of interest, that it works and is likely to be easily applicable.

The levels of GFP signal are disappointingly low for many genes, and not much information is provided on how they were imaged – only using a 20x objective in an LSM 880 confocal. If the authors could improve the brightness (or signal/noise ratio, which will have the same outcome), they would significantly improve the utility of their system and the insertions that are already available, e.g.:

- Presumably they are already using the maximum level of laser power that doesn't bleach their signal?

- Most 20x objectives are low NA – although they do not actually state the NA in their Materials and methods. Since fluorescence intensity in confocal or epifluorescence microscopy varies with the 4th power of NA, even small increases in NA can greatly improve S/N ratio or brightness, and oil or water objectives are far superior to air objectives for this. Even allowing for the 4-fold dimmer intensity of 40x compared to 20x, the higher NA of most 40x objectives also makes these images usually brighter, even if there is no scope to improve sensitivity with an available 20x objective.

- Using a larger pinhole setting will greatly improve brightness or S/N ratio of a weak signal since it increases the photon flux without a proportional increase in noise (albeit at the cost of z-resolution).

- Does lowering the laser scan speed, i.e. increasing pixel dwell time, increase the photons enough to improve the signal, if laser power cannot be increased further?

In conclusion I'm not convinced from their Materials and methods section that they've pushed sensitivity or S/N ratio to the limit. If not, I would like them to re-image their GFP-trap panels in Figure 3 and Figure 3—figure supplement 1 with acquisition settings that improve sensitivity and S/N ratio; if they have done this already, they need to document their efforts better and provide more details on the microscope acquisition settings.

---

## [Author Response]

Reviewer #1:[…] There is no doubt that DH is another extremely useful reagent developed by Bellen lab, and I do not have further comments. But it was not too clear to me whether the second method (two-step replacement of the gene) is significantly faster or more convenient than other already-applied CRISPR methods to generate endogenous tag. This method involves 1) generation of a gene specific construct to generate knock-out flies, and 2) generation of a construct that can tag or point-mutate the gene of interest, therefore need to be conducted at gene-by-gene basis. But I do see a point that DH and CRISPR methods are two complementary methods that are intended to cover all protein coding genes in the Drosophila genome, so it makes sense to report in a single paper. They have already done their best to explain the advantages of their CRISPR method, so I don't have additional suggestions, but I am just noting that I was a bit puzzled when they moved onto CRISPR method because two methods are very different.

We agree that the two methods are very different. However, the methods complement each other and facilitate manipulation of virtually all genes in the *Drosophila* genome. Also, the use of the *yellow* wing SIC, while not faster, is precise and reliable. The dominant marker is also more convenient than other CRISPR-based methods.

Reviewer #3:[…] The levels of GFP signal are disappointingly low for many genes, and not much information is provided on how they were imaged – only using a 20x objective in an LSM 880 confocal. If the authors could improve the brightness (or signal/noise ratio, which will have the same outcome), they would significantly improve the utility of their system and the insertions that are already available, e.g.:- Presumably they are already using the maximum level of laser power that doesn't bleach their signal?- Most 20x objectives are low NA – although they do not actually state the NA in their Materials and methods. Since fluorescence intensity in confocal or epifluorescence microscopy varies with the 4th power of NA, even small increases in NA can greatly improve S/N ratio or brightness, and oil or water objectives are far superior to air objectives for this. Even allowing for the 4-fold dimmer intensity of 40x compared to 20x, the higher NA of most 40x objectives also makes these images usually brighter, even if there is no scope to improve sensitivity with an available 20x objective.- Using a larger pinhole setting will greatly improve brightness or S/N ratio of a weak signal since it increases the photon flux without a proportional increase in noise (albeit at the cost of z-resolution).- Does lowering the laser scan speed, i.e. increasing pixel dwell time, increase the photons enough to improve the signal, if laser power cannot be increased further?

We agree that the GFP signal of the endogenously-tagged proteins is sometimes low, especially in the nervous system. We already reported this before. Note the expression in third instar larvae is still significantly stronger than in the adult brains. The noise ratio is difficult to optimize and often affects the quality of imaging (Diao et al., 2015; Lee et al., 2018). To address this issue we invested additional effort in optimizing the signal to noise ratio.In the previous version all the samples were imaged using rabbit anti-GFP directly fused to Alexa488 fluorophore. In this version we re-stained all the samples using rabbit anti-GFP and stained with a secondary antibody to amplify the signal. During the imaging we used many of the suggestions proposed by the reviewer and we were able to improve images for some. We therefore replaced images of larval brains for MI06872-CG34383, MI08614-Dgk and MI15073-CG9132 in Figure 3.

To document the usefulness of DH tagging with GFP we decided to dissect ovaries and image ovarioles. We focused on stage 9 and 10 egg chambers where different cells can easily be identified and it is relatively easy to assess subcellular protein distribution. We inserted a whole new figure (Figure 4) showing that it is pretty straightforward to stain and image with high resolution in ovarioles using similar settings that we used in the brain. We hope we have appeased the reviewer’s concern with these new data. Specifically, we added the following paragraph:

“As the GFP protein traps should be able to report the subcellular localization of the tagged protein we turned to tissues were subcellular localization and specific cell expression is easily assessed. […] In summary, GFP protein tagging with DH can be used to determine the cellular and subcellular localization of tagged proteins.”

To further highlight the usefulness of the GFP-tagged proteins, we also added the following text to the Discussion:

“for most genes the corresponding GFP-tagged protein signal in adult brains is often weak, consistent with previous results (Lee et al., 2018a; Diao et al., 2015). However, all the lines tested in the brain allow us to rapidly and reliably determine the cellular and subcellular localization of the GFP tagged proteins in egg chambers.”

We also included additional information in the Materials and methods section to describe our imaging settings with confocal microscopy:

**“**To increase signal, some samples used anti-GFP antibody (Invitrogen, A11122) followed by incubation with secondary antibody (Alexa Fluor 488-conjugated goat anti-rabbit IgG). […] Laser intensity and detector gains were adjusted as needed to increase signal-to-noise ratio and prevent signal saturation.”